# Variable temperature processing by plasmodesmata regulates robust bud dormancy release

Shashank K. Pandey[1,9], Tatiana S. Moraes[2,9], Aswin Nair[1,9], Bibek Aryal[1], Abdul Azeez [1,3], Pal Miskolczi [1], Guillaume Maucort [4], Fabrice P. Cordelières [4], Lysiane Brocard[4], Gwendolyn V. Davis [5], Hannah Dromiack [6,7], Swanand Khanapurkar [6,7], Sara I. Walker [6,8], George W. Bassel[5] ✉, Emmanuelle M. Bayer [2] ✉ & Rishikesh P. Bhalerao [1] ✉

Dormancy is a key mechanism in perennial plants in boreal and temperate regions, protecting buds from winter damage by repressing precocious bud break before spring onset. How plants robustly time dormancy release under fluctuating environments remains unknown. Here, we show that, rather than simply sensing cold duration, buds leverage warm spikes to sense winter progression and time dormancy release. This timing mechanism is mediated by previously unrecognized regulation of plasmodesmata by warm spikes acting through tree ortholog of *FLOWERING LOCUS T* (*FT1*) and the gibberellic acid pathway. Our results reveal FT1 as a previously unrecognized, suppressor of callose levels and show that warm spikes repress cold induction of FT1 and GA pathway to suppress PD opening and dormancy release. Importantly, buds exhibit heterogeneity in bud break. This heterogeneity in bud break crucial for bet hedging is amplified under temperature fluctuations and is associated with the thermal responsiveness of plasmodesmata. Altogether, our work reveals dynamic plasmodesmata regulation as a crucial tissue-level mediator of variable temperature processing by buds, enabling robust adaptation of trees to seasonal changes.

Boreal and temperate forest represent 45% of the world's forests[1], and experience massive seasonal shifts in temperature throughout the year[2,3]. To survive these fluctuations, perennial plants such as long-lived trees halt growth and establish dormancy prior to winter[4]. The growth is reinitiated at the onset of spring, thereby synchronizing growth and dormancy cycles with seasonal changes. Temporal regulation of these cycles is vital for ecosystem function, impacting carbon sequestration[5,6], synchronization of tree phenology with that of animals[7,8], and forest productivity[9].

In autumn, shoot apical meristem (SAM) activity in perennial trees is suppressed, halting leaf primordia formation and growth, and SAM and leaf primordia are enclosed within apical buds[10]. Following growth cessation, bud dormancy is established[11]. Dormancy suppresses reactivation of growth in the buds (bud break) until the onset of spring,

¹Umeå Plant Science Centre, Department of Forest Genetics and Plant Physiology, SwedishUniversity of Agricultural Sciences, Umeå, Sweden. ²Laboratoire de Biogenèse Membranaire, CNRS, Université de Bordeaux, Ornon, France. ³Powerpollen, Ames, IA, USA. ⁴Bordeaux Imaging Center, UAR 3420 CNRS, Université de Bordeaux -US4 INSERM, Bordeaux, France. ⁵School of Life Sciences, University of Warwick, Coventry, UK. ⁶Beyond Center for Fundamental Concepts in Science, Arizona State University, Tempe, AZ, USA. ⁷Department of Physics, Arizona State University, Tempe, AZ, USA. ⁸School of Earth and Space Exploration, Arizona State University, Tempe, AZ, USA. ⁹These authors contributed equally: Shashank K. Pandey, Tatiana S. Moraes, Aswin Nair. ✉e-mail: gbassel@gmail.com; emmanuelle.bayer@u-bordeaux.fr; Rishi.Bhalerao@slu.se

thereby protecting them from freezing-induced damage during winter[12]. Premature release of bud dormancy can result in precocious bud break, risking the exposure of buds to extreme low temperatures, causing irreversible damage[13–16]. Robust temporal regulation of seasonal growth cycles is therefore crucial for tree survival, but perennial trees must also be able to respond to variable climatic parameters.

Prolonged low temperatures (4–8 °C) serves as environmental cues sensed by buds to trigger dormancy release upon reaching a chilling threshold[17]. The antagonistically acting plant hormones abscisic acid (ABA) and gibberellic acid (GA), along with the tree ortholog of the *Arabidopsis* florigen component *FLOWERING LOCUS T* (*FT1*) play a key role in bud dormancy[18–25]. ABA suppresses cell-to-cell communication by deposition of callose, blocking plasmodesmata (PD) to induce bud dormancy[21,25–29]. Conversely, low temperatures trigger dormancy release by inducing the expression of *FT1*, MADS-box transcription factor *LIM1* and GA biosynthesis related genes such *GA20-oxidase*[22–24,30].

In nature, daily temperatures can fluctuate between optimal cold interrupted by warm spikes. Consequently, to reliably achieve chilling threshold, buds must distinguish temporary (cold interrupted by long warm spikes) from prolonged (cold interrupted by shorter or a complete absence of warm spikes) cold. While key components of the genetic networks regulating dormancy release have been uncovered[4,31], how the buds leverage these to reliably time dormancy release under naturally variable temperature regimes remains enigmatic[4,32].

Using hybrid aspen trees, we show that dormant buds do not rely solely on measurement of cold duration but also on the presence and duration of warm spikes to reliably sense the passage of winter and robustly time dormancy release. This mechanism highlighting the previously uncharacterized role of warm spikes, enables buds to distinguish intermittent from prolonged cold and optimally time dormancy release even under variable cold temperatures. This variable temperature processing is mediated by the dynamic response of PD regulated by FT1 and GA. Intriguingly, PD-mediated control of the cell–cell network may also contribute to heterogeneity in bud break crucial for bet-hedging, a survival mechanism for adapting to variable environments, during bud break. This dynamic control of cell–cell communication serves as a previously unrecognized strategy used by trees to adapt and respond effectively to unpredictable environmental changes, thereby enhancing their survival.

## Results

### Buds process variable temperatures to regulate dormancy release

Trees typically experience cold temperature fluctuations with intermittent warm spikes in nature. It remains unclear how buds process these variable temperature signatures to regulate dormancy release. To address this, we investigated dormancy release by treating hybrid aspen (clone T89) dormant buds to constant cold or simulated temperature fluctuations incorporating low temperatures interspersed with warmer spikes (Fig. 1a). We used bud break following the two temperature treatments of dormant buds to assess dormancy release (as bud break cannot occur until dormancy is released[21]. The hybrid aspen clone T89 (WT) plants, were exposed to 11 weeks of short days (11WSD) to induce dormancy[21] followed by exposure to either constant cold (24 h at 4 °C) or variable cold (20 h at 4 °C/4 h at 20 °C) (Fig. 1a). We compensated for the reduced hours of cold within the variable cold regime by extending the duration of that treatment, so all buds received equivalent hours of cold (Fig. 1a). Following cold exposure, the trees were subjected to warm (22 °C) long days (LD; 18 h day/6 h night) to assay bud break. Consistent with earlier results, 100% bud break was observed in dormant buds exposed to 4 weeks of constant cold. In contrast, none of the buds exposed to variable cold underwent bud break, indicating that dormancy had not been released (Fig. 1b, c). Thus, prolonged cold interruption by warm spikes suppresses

dormancy release, even if buds receive equivalent cold hours as those exposed to constant cold. This shows that quantitative accumulation of cold hours alone does not account for temperature regulation of dormancy release. Importantly, buds can distinguish between continuous and fluctuating cold, indicating a mechanism for processing variable temperatures to facilitate robust timing of dormancy release in nature.

### PD respond dynamically to temperature fluctuations

We next investigated why buds exposed to variable cold failed to break dormancy despite receiving the same total number of cold hours as those exposed to constant cold (Fig. 1a). The blockage of PD by callose is essential for dormancy induction[21,26,27]. Conversely, low temperature-induced callose reduction, facilitating PD opening, is associated with dormancy release. Therefore, we quantified the callose levels at PD via immunolocalization with callose-specific monoclonal antibody[33], in 11WSD dormant buds and after four weeks of either constant or variable cold exposure. In agreement with prior studies, callose levels decreased in dormant buds exposed to constant cold (Fig. 2a, Supplementary Fig. 1a, Supplementary Fig. 2a). However, no such reduction in callose was observed after exposure of buds to variable cold (Fig. 2a, Supplementary Fig. 1a, Supplementary Fig. 2a). Spatial analysis of PD-associated callose levels at a single-cell level showed an increase in the proportion of cells with low callose when buds were exposed to constant cold. This shift was not observed in buds exposed to variable cold (which also failed to trigger dormancy release) (Fig. 2b and Supplementary Fig. 1b–c, Supplementary Fig. 2b). As the abundance and distribution of PD remained unchanged under constant and variable cold (Supplementary Fig. 3), temperature processing appears to act via callose-mediated control of PD opening, not their density. These results showing that PD open in response to constant but not variable cold, point to the crucial role of PD in variable temperature processing in dormancy release.

### Constitutively open PD compromise variable temperature processing

As opening of PD is suppressed under variable cold, we investigated whether PD dynamics contributes to temperature processing during dormancy release. To test this, we uncoupled PD opening from its regulation by temperature. If PD gating is important for dormancy release, this should override the inhibitory effect of warm spikes on dormancy release. For testing this, we used *abi1* mutant in which PD are constitutively open. In previous work, we have shown that ABA induces PD closure, and when ABA response is suppressed, as in *abi1* mutant, callose levels are highly reduced (Supplementary Fig. 4). Consequently in *abi1* mutant PD remain constitutively open[21]. Importantly, overexpression of PDLP, a PD-specific regulator that reduces trafficking via PD, fully suppresses *abi1* dormancy defects (without affecting ABA response due to expression of *abi1-1*), showing that ABA regulates dormancy through PD closure by inducing callose accumulation in buds[21].

We exposed *abi1-1* plants and wild-type (WT) control plants to 11WSD, followed by exposure to either constant or variable cold (20 h at 4 °C/4 h at 20 °C) (Fig. 1a), and then assessed bud break. Both WT and *abi1-1* buds broke dormancy under constant cold (Fig. 2c, d), but under variable cold only *abi1-1* buds underwent bud break (Fig. 2c, d). Thus, uncoupling PD regulation dynamics from temperature control result in bud break even under unfavorable variable cold conditions. Altogether these data indicate that the regulation of callose-mediated PD opening plays a crucial role in variable temperature processing and robust control of dormancy release.

### FT1 mediated control of PD opening is crucial for variable temperature processing

We next investigated upstream regulators of cold regulated dormancy release. Transcription factors *SVL* and *LIM1* have been reported as bud dormancy and PD regulators[24,27,34,35]. *SVL* promotes PD closure and

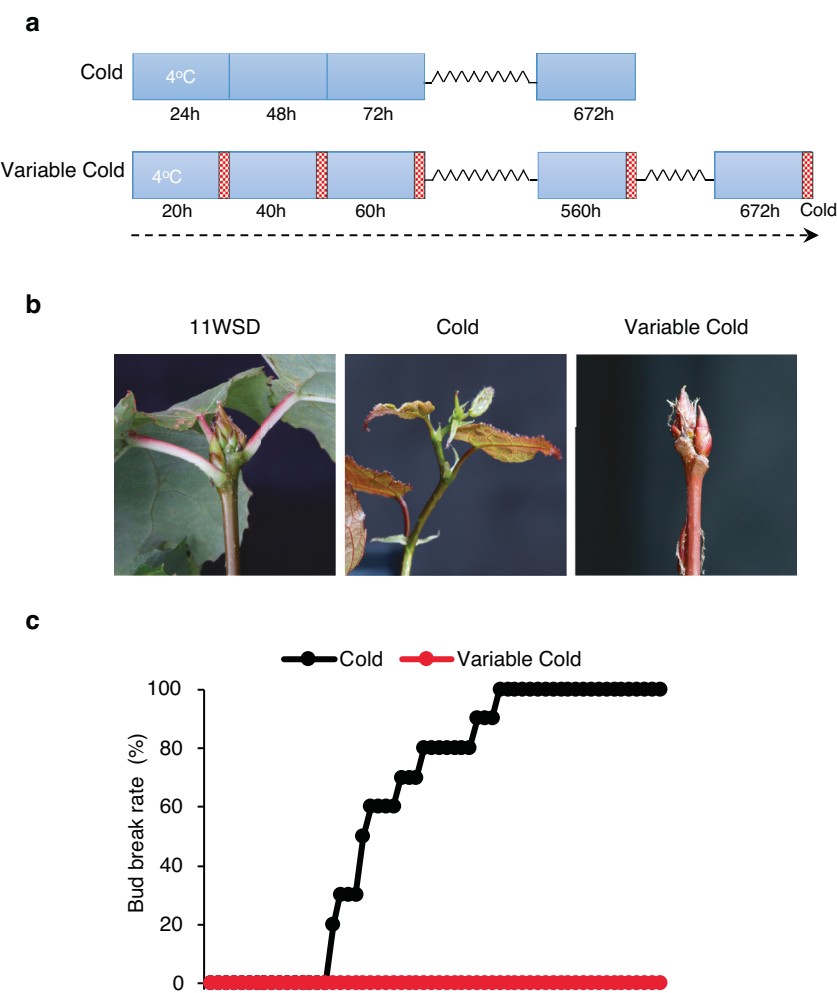

**Fig. 1 | Temperature fluctuations modulate bud dormancy release in hybrid Aspen. a** Schematic diagram showing the cold and variable cold treatment schedules. After 11 weeks of short days (SD), buds were exposed to cold. The cold treatment consists of continuous exposure to 4 °C for 4 weeks (672 cold hours) and the variable cold treatment involves a daily cycle of 20 h of cold exposure (4 °C), followed by 4 h of warm periods (20 °C) indicated by red cross-hatched boxes. The variable cold cycle was continued for additional time to ensure a total cold exposure of 672 h (equivalent to that for 4 weeks of constant cold). **b** Dormancy release in buds assayed by percent bud break following exposure of buds to constant or variable cold after 11 weeks of SD and (**c**) The percent bud break rate of hybrid Aspen (T89) plants in constant vs. variable cold temperatures. The experiments were repeated at least twice with similar results, and the bud-break rate (%) is shown with data from 10 to 12 plants.

positively regulates dormancy, whereas *LIM1* acting antagonistically, downregulates callose level to release dormancy. Importantly, dormancy release involves cold mediated downregulation of *SVL* expression and simultaneously upregulation of *LIM1*. These results prompted us to investigate their expression in response to variable cold compared with constant cold in buds. Our results show that *SVL* was similarly downregulated in buds exposed to constant or variable cold (Supplementary Fig. 5a). While *LIM1* was upregulated in both constant and variable cold with slightly reduced (statistically not significant) induction in variable cold (Supplementary Fig. 5b). These results showing that *SVL/LIM1* expression responds similarly to constant and variable cold suggest that change in *SVL/LIM1* expression is unlikely to be the driver of PD-mediated temperature processing.

We then focused on FT1, a cold induced promoter of dormancy release whose loss-of-function impairs dormancy release[22,30]. To assess FT1's role in variable temperature processing, we first analyzed its expression in dormant buds under constant and variable cold (Fig. 3a). Upon exposure to constant cold, *FT1* expression was upregulated in the hybrid aspen buds, whereas FT1 expression remained significantly

low under variable cold conditions. These results suggest that failure to trigger dormancy release under variable cold is independent of transcriptional regulation of *SVL* or *LIM1* but is instead attributable to a failure to induce FT1.

Although FT1 is implicated in dormancy release[22], the mechanism by which FT1 contributes to dormancy release remains unknown. Interestingly, our grafting assay (Supplementary Fig. 6), showed that cold treated *ft1* buds failed to undergo bud break even when grafted on FT1 overexpressor root stocks (Supplementary Fig. 6). These data strongly hint at PD being closed in *ft1* mutant buds. However, in prior analysis of *ft1* mutant[22], callose levels were not analysed and therefore, it is not known whether FT1 promotes dormancy release by directly acting on PD e.g., facilitating PD opening by downregulating callose. Since FT1 expression remains low under variable cold when callose levels are high and dormancy release is suppressed, we examined FT1's function in temperature-driven PD regulation and dormancy release.

We first tested if FT1 is essential for cold induced downregulation of callose levels in the buds. For this we used a loss of function *ft1* mutant and analysed callose levels in buds before and after cold

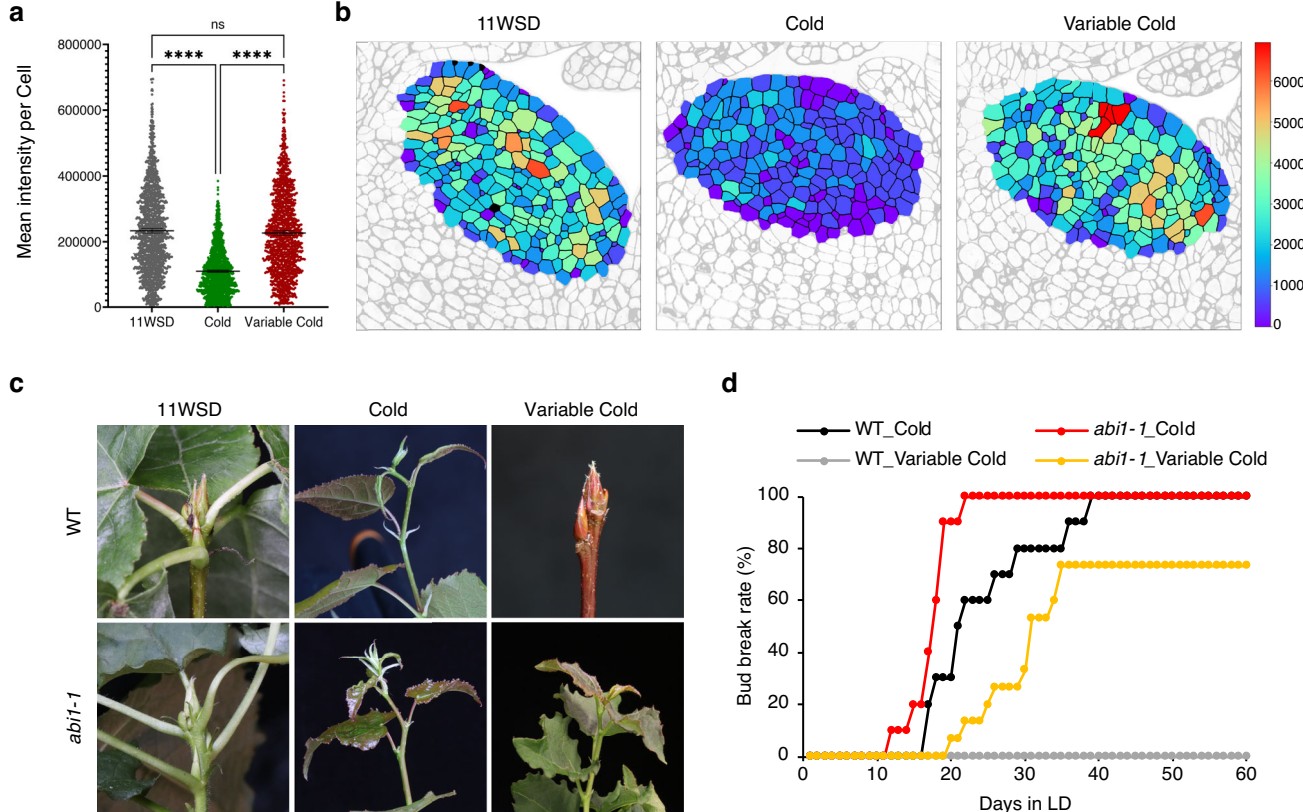

**Fig. 2 | Regulation of PD network is crucial for variable cold processing by buds.**
**a** Quantification and spatial distribution of callose at the cell level in the shoot apical meristem of aspen buds, after 11 weeks of short days and in response to 4 weeks of constant and variable cold temperatures. Graph displaying callose quantification per cell based on the sum of the mean signal intensities at the cell periphery (data points represent individual cells). Quantification was based on six meristem surfaces per condition, each with an area of 7515 μm², containing 220–280 cells. Statistical analysis was performed using one-way ANOVA ($p < 0.0001$) followed by Tukey's test. Error bars represent the 95% confidence interval of the difference, and asterisks (****) denote a significant difference.

**b** Visual representation of PD-associated callose levels per cell. The color legend bar indicates the sum of the callose mean signal intensities at the cell periphery.
**c** Response of WT and *abi1-1* buds to constant and variable cold temperatures. WT and *abi1-1* plants grown under short-day (SD) conditions for 11 weeks were exposed to either 4 weeks of constant cold or variable cold temperatures (as in Fig. 1a) and then transferred to warm, long-days, and bud break was recorded. **d** The percent bud break of WT and *abi1-1* plants subjected to constant cold versus variable cold temperatures. The experiment was repeated at least twice with similar results, and the bud-break percentage is shown with data from 10–15 plants.

treatment (Fig. 3b,c) with high-resolution imaging using anti-callose immunogold labeling and electron microscopy. Unlike wild-type buds, where cold triggered callose downregulation, *ft1* mutant buds showed no such reduction. Importantly, the PD density in the buds of *ft1* mutant was not significantly different between 11WSD and constant cold, as also observed in the wild type (Supplementary Figs. 3 and 7). These results suggest that FT1 is required for PD opening via callose downregulation in response to cold. Moreover, *ft1* mutant was not able to undergo dormancy release even after exposure to constant cold (or variable cold with 4 h warm spikes) (Fig. 3g and Supplementary Fig. 11a,b). These results highlight the requirement of FT1 for callose downregulation and dormancy release in response to cold.

Next, we investigated if FT1 is sufficient to induce callose downregulation. To test this, we generated transgenic plants in which FT1 could be induced chemically: hybrid aspen plants expressing FT1 were fused with green fluorescent protein (GFP) and hemaglutinin (HA) tags under the control of a steroid (estradiol)-inducible promoter (XVE::FT1-GFP-HA). The XVE::FT1-GFP-HA and WT control plants were exposed to 11WSD to induce bud dormancy. The efficiency of the inducible system was validated by investigating *FT1* expression before and after estradiol treatment in the dormant buds, using DMSO (mock control) or 10 μM estradiol (Supplementary Fig. 8), and continuing to keep them under non-inductive short days without exposing them to cold-temperature. Dormant wild-type buds did not show any

detectable expression of *FT1* before or after treatment, while, in XVE::FT1-GFP-HA plants, *FT1* expression was negligible in mock-treated dormant buds but significantly induced after 2 weeks of estradiol treatment (Supplementary Fig. 8).

We then assessed if inducing *FT1* expression could reduce PD callose levels using again anti-callose immunogold labeling and electron microscopy. Mock and estradiol treatments had no effect on callose levels in wild-type buds. However, in XVE::FT1-GFP-HA plants, estradiol induced FT1 resulted in significantly reduced callose levels compared to mock-treated buds (Fig. 3d,e). Furthermore, gene expression analysis showed that FT1 regulates the expression of key genes of GA pathway that is known to promote callose downregulation[24]. FT1 induction promotes the expression of GA20-oxidase, encoding the key enzyme in GA biosynthesis pathway and suppressed the expression of catabolism related GA2-oxidase (Supplementary Fig. 9). Moreover, FT1 induction also modulated the expression of PD-related markers, upregulating the expression β-1,3-GLUCANASE 1 (BG1) and downregulating PDLP6-2, and MCTP7 expression (Supplementary Fig. 10). These results together with the failure of *ft1* mutant to downregulate callose levels in response to cold, highlights the link between FT1 and PD regulation and shows that FT1 upregulation is necessary and sufficient to downregulate callose level.

Since PD opening is crucial for dormancy release (and FT1 facilitates PD opening), we then investigated whether FT1 induction is

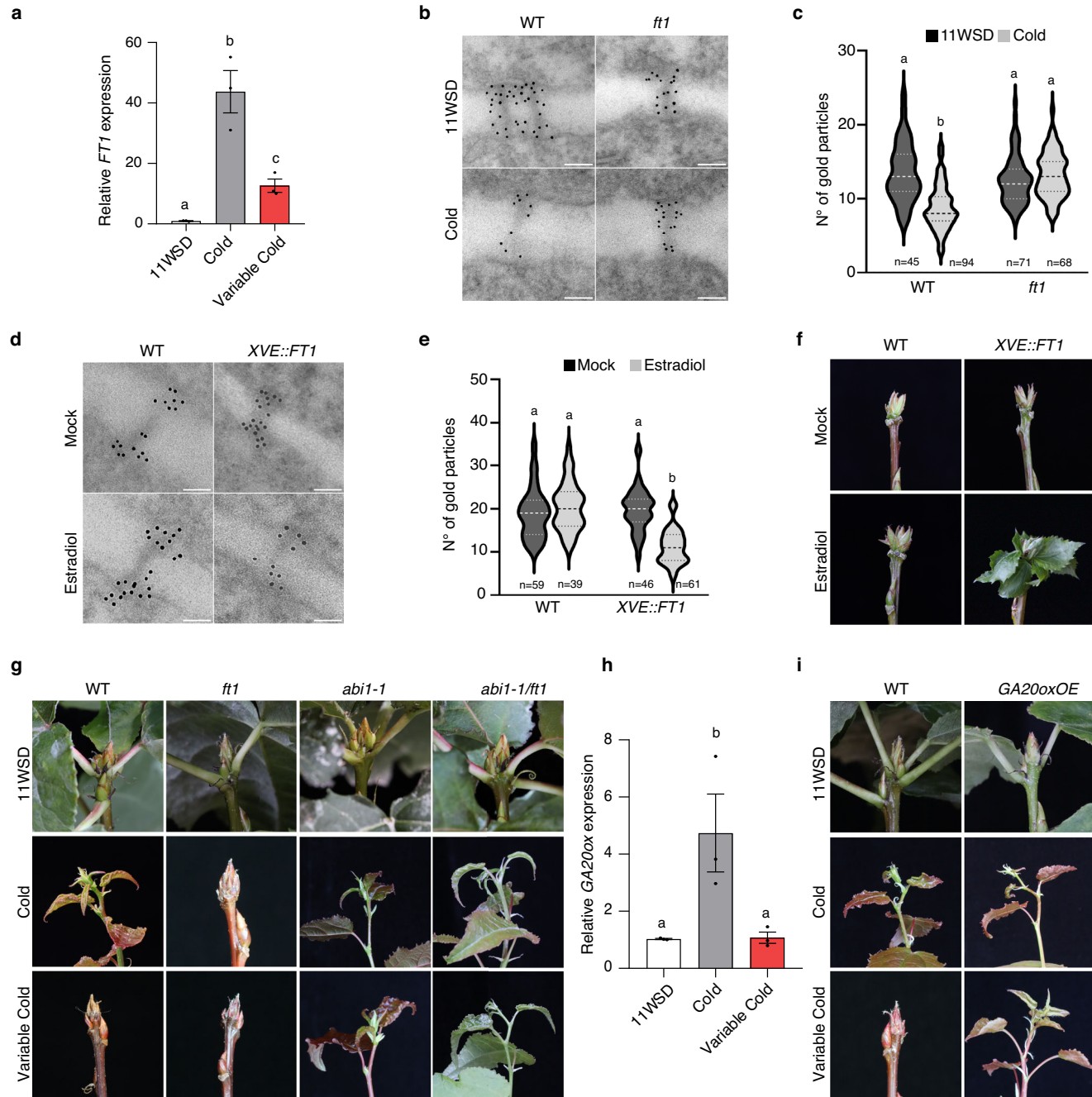

**Fig. 3 | FT1 and GA mediate the variable cold response of buds. a** Relative expression of *FT1* in wild-type buds after 11 weeks of short-day (SD), followed by 4 weeks of constant cold and variable cold temperatures. Expression values shown are normalized to the reference gene *UBQ* and are averages of three biological replicates. Each biological replicate consists of three buds. Error bars indicate standard error mean (± SEM). Statistical significance was determined using one-way ANOVA and multiple comparison by the Holm−Sidak method ($P < 0.05$). Letters above the bars indicate significant differences. **b**, **d** High-resolution TEM images of PD ultrastructure and anti-callose labeling (10 nm gold particles) in aspen buds. **b** WT and FT1crispr (*ft1*) after 11 weeks of SD and 4 weeks of cold. **d** WT and FT1 inducible line (*XVE::FT1-GFP-HA*) under mock and estradiol treatment. Scale bars = 0.1 μm. Plants were shifted to SD conditions for 11 weeks to induce bud formation. Estradiol treatment was done in the SD condition and samples were collected after 2 weeks for TEM analysis. **c**, **e** Callose quantification based on the number of gold particles in PDs.

Different letters indicate significant differences based on a one-way ANOVA ($p < 0.0001$) followed by Tukey's test. **f** Bud break analysis in WT and XVE::FT1-GFP-HA line treated with mock or estradiol. **g** Response of WT, *ft1*, *abi1-1* and *abi1-1/ft1* buds exposed to constant vs. variable cold temperatures. Plants were grown under short-day (SD) conditions for 11 weeks, followed by either 4 weeks of constant cold or variable cold temperatures (as described in Fig. 1a) then transferred to warm, long-days, and bud break was recorded. **h** Relative expression of *GA20ox* in wild-type buds after 11 weeks of SD, followed by 4 weeks of constant cold and variable cold temperatures. Expression values shown are normalized to the reference gene *UBQ* and are averages of three biological replicates, each consisting of three buds. Error bars indicate standard error mean (± SEM). Statistical significance was determined as indicated in Fig. 3a. **i** Response of WT and *GA20oxOE* buds exposed to constant vs. variable cold temperatures. Plants were grown constant and variable cold conditions as described in Fig. 3g.

sufficient to induce dormancy release even in the absence of constant cold temperature using FT1-inducible line. Wild-type and XVE::FT1-GFP-HA plants were exposed to 11WSD to induce bud dormancy. Following this, buds of both wild-type and the inducible lines were exposed to DMSO (mock) or estradiol and assessed for bud break under non-inductive short days without cold treatment (Fig. 3f, Supplementary Fig. 8). Neither mock nor estradiol treatment resulted in bud break in wild-type buds. Similarly, mock-treated XVE::FT1 buds also maintained their dormancy, whereas estradiol-treatment triggered bud break (Fig. 3f, Supplementary Fig. 8). Thus FT1 induction is sufficient and can substitute for cold in inducing PD opening and dormancy release.

These results show that (1) variable cold does not act via *SVL* or *LIM1*, (2) *FT1* expression responds to different temperature regimes, being induced by constant cold but not by variable cold, and (3) *FT1* expression promotes PD opening by reducing callose levels and facilitating dormancy release. Our results thus establish a previously unknown link between the control of PD regulation by FT1. This suggests a scenario where temperature processing is integrated through *FT1* expression with variable cold suppressing dormancy release by repressing *FT1* induction and antagonizing PD opening.

To validate this further, we generated a double *abi1-1/ft1* mutant and tested its response to variable cold. Since PD remain constitutively open in *abi1-1* due to reduced ABA response, FT1 should no longer be required for cold-induced PD opening in this mutant. In other words, if temperature processing relies on FT1 to open PD, FT1 should become dispensable once PD are already open. Indeed, our data showed that, in contrast to wild type, *abi1-1/ft1* is insensitive to variable cold and undergo bud break (Fig. 3g, Supplementary Fig. 11a–d). Altogether, these results indicate that warm spikes act via *FT1* to suppress downregulation of callose which then prevents dormancy release.

### Variable cold suppresses GA pathway a regulator of PD

GA (like FT1) is a promoter of dormancy release and has been shown to promote callose reduction at PD in response to cold[24,36]. For example, hybrid aspen plants overexpressing GA20-oxidase (high GA levels) have low callose and conversely, GA2-oxidase (low GA levels) have high callose levels compared with wild type buds[24]. Importantly, reducing GA levels, like overexpression of PDLP1, restores dormancy in *abi1-1* mutant[35] and conversely, exogenous GA application is sufficient to induce callose downregulation and dormancy release[23]. We first checked if the expression of *GA20-oxidase 1*, a key enzyme in GA biosynthesis, was, like FT1, differentially regulated by constant or variable cold. Data showed that *GA20-oxidase 1* expression mirrored *FT1*, being upregulated under constant cold but not under variable cold (Fig. 3h). These results suggest that under variable cold, failure to induce GA20-oxidase, may suppress dormancy release (as for FT1). To test this, we used transgenic hybrid aspen plants with elevated GA levels (by constitutive overexpression of *GA20-oxidase*)[37] to uncouple the regulation of the GA pathway from temperature fluctuations. We subjected wild-type and GA20-oxidase overexpressors (GA20oxOE) to 11WSD, followed by constant or variable cold, and then investigated dormancy release (Fig. 3i, Supplementary Fig. 11a,e). Unlike WT, which do not break dormancy, nearly 50% of GA20oxOE plants underwent bud break under variable cold. Thus, uncoupling the GA pathway from temperature regulation compromises buds from processing variable temperature. Altogether, these data indicate that variable temperature processing is mediated by GA together with FT1, two key regulators of PD opening.

### Heterogeneity in bud break is enhanced under temperature fluctuations and attenuated in *abi1-1* mutant

At a population level, species can mitigate against the impacts of variable and unpredictable environments by generation of heterogeneous responses typically described as bet hedging[38]. For example,

in genetically identical bacterial populations, a small fraction of slow-growing cells enables survival under antibiotics[39–41]. Bet-hedging has also been reported in plants[42–44] but, the underlying mechanisms remain poorly understood.

Since variable responses can protect against unpredictable future conditions, we compared if bud break timing varies between WT buds exposed to both constant and variable cold. We used 2-h warm spikes instead of 4-h warm spikes, as the latter treatment completely blocked dormancy release (Fig. 4a). In contrast with 4-h warm spikes, WT buds were able to undergo bud break when exposed to cold with intermittent 2-h warm spikes (Fig. 4b, Supplementary Fig. 12) and importantly, this corresponded with a significant reduction in PD-associated callose levels in the buds (Supplementary Fig. 13).

Our data showed a substantial variation in bud break timing across populations of genetically identical WT hybrid aspen buds (Fig. 4b). Intriguingly, this heterogeneity in bud break, crucial for bet-hedging, was significantly reduced in *abi1-1* trees compared to WT (cov = 1.2 in *abi1-1* vs. cov = 18.72 in WT). Interestingly, variable cold significantly increased the variation in timing of bud break in WT (constant cold cov =18.72 vs. variable cold cov = 39.76), indicating a strong enhancement of this response under variable cold compared to constant cold (Fig. 4b, c). In contrast with the wild type, in the *abi1-1* mutant, the variation in timing of bud break was reduced compared to WT under both constant (cov = 1.2 in *abi1-1* vs. cov = 18.72 in WT) and variable cold (*abi1-1* cov = 2.2 vs. WT cov = 39.76, suggesting that the mechanism contributing to variability in bud break is compromised in the *abi1-1* mutant with open PD. Our results thus show that buds display a heterogeneous response in bud break typically associated with bet hedging. This heterogeneity is enhanced in response to variable temperatures and this response is strongly attenuated in *abi1-1* mutant compromised in regulation of PD and variable temperature processing.

## Discussion

Long-term integration of variable temperatures is crucial for robust temporal regulation of several developmental transitions in plants, such as seasonal growth cycles[4,32,45–48] and flowering. Temperature is an inherently noisy cue and it is unclear how plants perform long-term integration of variable temperature to reliably time these crucial developmental transitions[49].

Cold interruption by warm spikes influences developmental transitions in other contexts with opposing effects. It promotes seed dormancy release[50] but inhibits cold registration, suppressing vernalization[51]. Temperature integration and vernalization suppression is mediated via epigenetic regulation[51]. Here, we reveal that plants use PD-mediated cell–cell communication as a multiscale integrator of variable temperature processing through the detection of warm spikes, to ensure accurate bud dormancy release and bet-hedging.

Buds rely on long-term exposure to cold to sense the winter transition and time dormancy release thereby synchronizing bud break with the advent of spring[4,32]. Dormancy release relies on progressive accumulation of chilling. Therefore, the timing of when low temperature registration begins is critical determinant of robust dormancy release. During the transition from late summer/autumn to winter, temperatures decrease, but not uniformly, with intermittent warm spikes interrupting the daily, potentially dormancy-breaking, cold periods.

Here we address how buds integrate cold when daily temperature fluctuates with warm spikes, to reliably time dormancy release. By investigating dormancy release under fluctuating cold, our results reveal an unexpected role for these warm spikes in variable temperature processing by buds for robust timing of dormancy release. Long warm spikes block dormancy release by suppressing cold integration; conversely, with shorter warm spikes, cold is registered and dormancy can be broken. This differential sensitivity of buds to longer vs

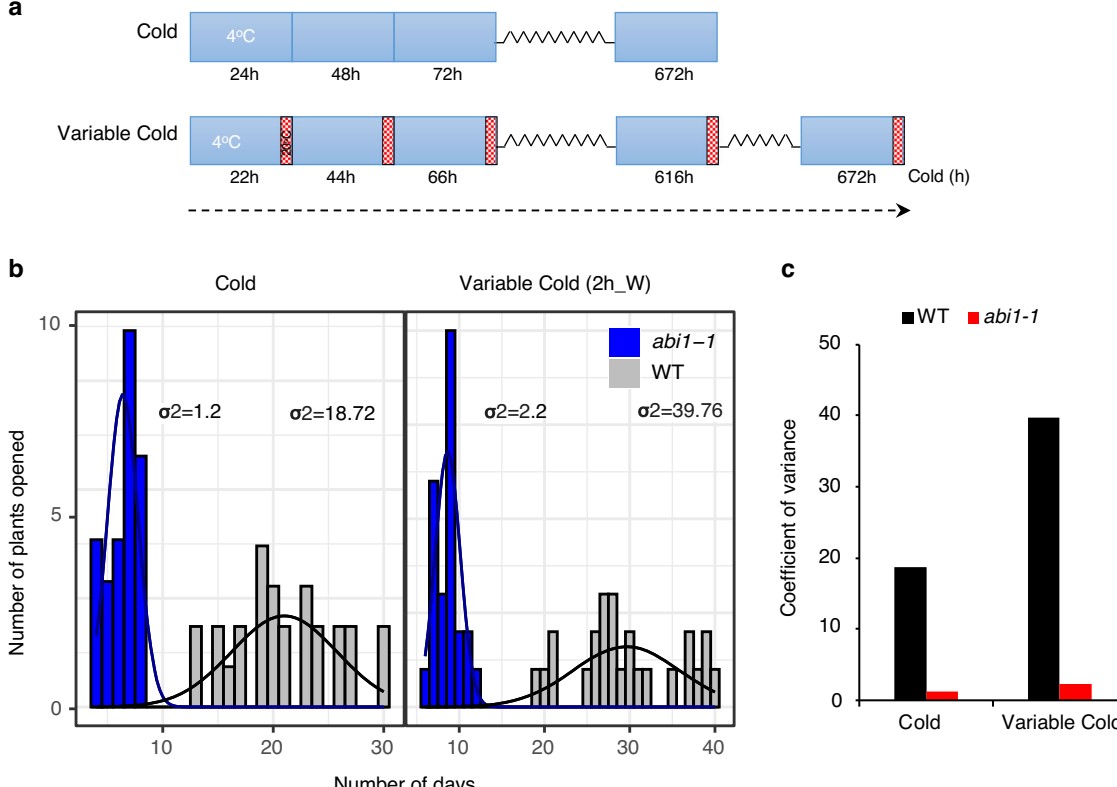

**Fig. 4 | The bet-hedging response of buds is altered in *abi1-1* mutant.**
**a** Schematic diagram showing the cold and variable cold treatment schedules. The cold treatment consisted of a continuous exposure to 4 °C for 4 weeks (672 h) and the variable cold treatment involved a daily cycle of 22 h of cold (4 °C) exposure followed by 2 h of 20 °C warm period indicated by red cross-hatched boxes. The variable cold treatment was extended to ensure buds received equivalent cold exposure of 672 h. **b** The histogram graph shows the distribution of the days required for bud break under constant versus variable temperature conditions for WT (gray bars) and *abi1-1* (blue bars). The solid lines overlaid on the histograms represent the distribution of plant opening events over time for each genotype. In the left panel, bud break timing in plants subjected to constant cold (24 h at 4 °C) is shown and the variability in the bud break timing is indicated by the variance values

($\sigma^2$) for each genotype. The *abi1-1* mutant shows lower variance ($\sigma^2 = 1.2$) compared to WT ($\sigma^2 = 18.72$), suggesting reduced heterogeneity in bud break in *abi1-1* exposed to constant cold. In the right panel, bud break timing in WT and *abi1-1* mutant exposed to variable cold (22 hr at 4 °C and 2 h at 20 °C) is shown. The *abi1-1* mutant exhibits a lower variance ($\sigma^2 = 2.2$) compared to WT ($\sigma^2 = 38.76$), indicating a more synchronized bud break response over time. **c** Bar graphs represent the coefficient of variation (CV) as a measure of heterogeneity in timing of bud break. Under constant cold, *abi1-1* shows a significantly lower CV compared to WT, reflecting a reduced heterogeneity in bud break. Similarly, under variable cold, *abi1-1* mutant maintains lower CV, further indicating reduced heterogeneity in response to variable temperatures.

relatively shorter warm spikes underpins a cold integration mechanism that enables robust sensing of autumn to late autumn/winter transition, as, typically, there is a shift towards progressively shorter periods of warm spikes relative to cold. A lack of response to cold when the warm spikes last longer can therefore ensure that cold registration is not initiated and chilling threshold is not achieved prematurely, preventing precocious dormancy release (Fig. 5). The sensing of warm spikes can thus contribute to a robust yet flexible means of responding to seasonal transitions despite the noisy temperature signal. Importantly, our results show that chilling fulfillment is more complex than simply being a cumulative process; accounting for warm spikes enables the essential processing of variable temperatures needed for dormancy regulation.

Dormancy release is associated with cold mediated reopening of PD, which restores cell-cell communication through callose downregulation[23,24,26]. These results link chilling accumulation with timing of dormancy release via PD regulation. However these earlier findings are based on exposing buds to constant cold and have not addressed how dormancy release is regulated under fluctuating cold interspersed with warm spikes. Our results address this scenario by revealing that warm-spikes prevent dormancy release and show that the integration of warm spikes occurs at the level of cell-cell communication such that in presence of warm spikes by PD remain closed

instead of reopening, despite buds receiving necessary total cold hours required for PD opening. Cold promotes dormancy release by suppressing expression of *SVL*, a promoter of PD closure and simultaneously inducing *LIM1* that triggers PD opening. However, we show that, intriguingly, warm spikes do not suppress *FT1* expression by targeting *SVL* or *LIM1*, the antagonistic acting upstream regulators of *FT1*. These results therefore suggest a previously undescribed mechanism for warm spike regulation of *FT1* distinct from the canonical cold regulation of *FT1* in buds crucial for variable temperature processing.

Notably, our findings reveal a role of *FT1* in callose downregulation, identifying FT1 as a key temperature-sensitive inducer of PD opening. This previously unrecognized role of *FT1* in PD opening thus explains how *FT1* contributes to dormancy release that was not well understood. Importantly, these results strongly suggest that suppression of dormancy release by warm spikes is due to their repression of FT1 induction which then results in a failure to open PDs. Warm spikes also suppress the expression of *GA20-oxidase*, key enzyme of GA biosynthetic pathway that promotes dormancy release and PD opening[24]. Consequently, ectopically enhancing GA levels (in GA20oxOE), to uncouple GA response from cold regulation, interferes with variable temperature processing by buds. Interestingly, in GA20oxOE plants, enhancement of GA levels reduces but does not

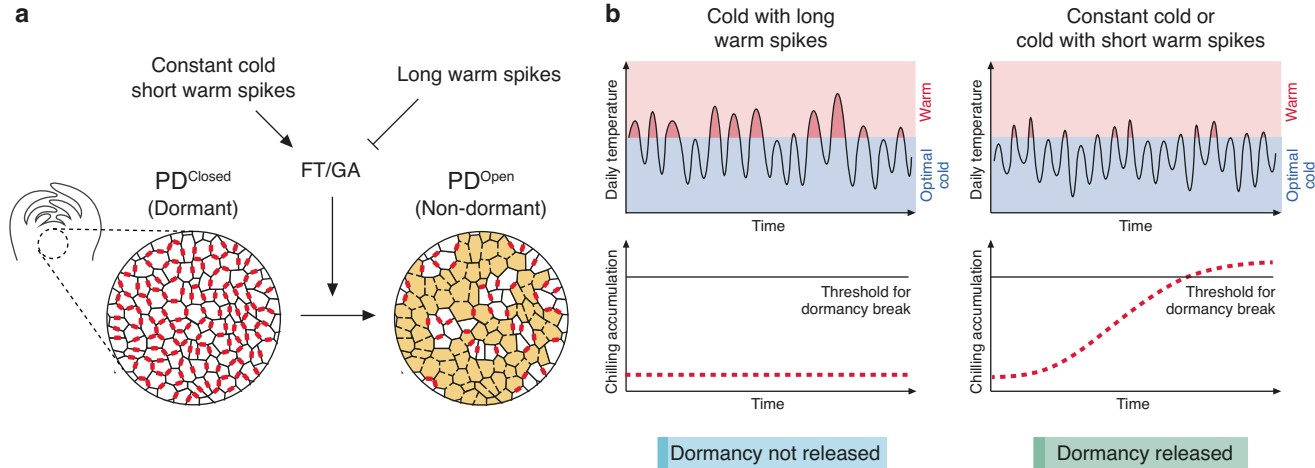

**Fig. 5 | A schematic model for variable temperature processing for regulation of bud dormancy release by buds. a** Temperature regulation of PD dynamics. A cross-section of a bud with cells is highlighted. In dormant buds, callose (red dots) block PD and suppress cell-cell communication (on the left). Upon exposure of the dormant buds to constant or cold interrupted with short warm spikes, the expression of *FT1* and GA pathway is activated, inducing the removal of callose from PDs. Resultantly, the proportion of cells lacking callose at PDs increases (highlighted in yellow) facilitating cell-cell communication and triggering dormancy release. Contrastingly, in the variable cold with long warm spikes (shown as block arrow), activation of FT1/GA is suppressed and consequently, callose levels cannot be downregulated, PDs cannot be unblocked and dormancy release is suppressed. **b** When buds experience cold with long warm spikes (e.g., in early autumn), cold registration (shown as red dashed line) is suppressed and chilling accumulation is not initiated and buds do not achieve the threshold for dormancy release. However, as buds start to experience either constant cold or cold with shorter warm spikes (later in the year), buds initiate cold registration gradually and release dormancy once the chilling threshold is achieved. This mechanism prevents premature initiation of cold registration and precocious dormancy release ensuring alignment with seasonal changes.

completely suppress the sensitivity of buds to warm spikes indicating that additional factors are also involved in the process. Our data thus support a scenario where downregulation of callose required for opening of PDs is suppressed due to repression of FT1 and GA pathway under variable cold. Together these findings highlight the critical role of PD-mediated cell-cell communication via FT1-GA pathway in variable temperature processing and ensuring robust temporal control of dormancy release.

Interestingly, Murata et al.[52] showed that in Arabidopsis, cold induced ABA restricts FT1 movement by promoting closure of PD. These results are in agreement with our findings showing that ABA is required for PD closure via upregulation of callose levels in hybrid aspen buds[21]. Thus, ABA plays a conserved role in promoting closure of PD. Our data here indicate that cold-induced FT1 promotes PD opening by reducing callose deposition. This raises the question of whether the FT ortholog in Arabidopsis has a similar function, acting antagonistically to ABA, forming a regulatory loop controlling PD dynamics.

In addition to revealing a crucial role of variable temperature processing in robust timing of dormancy release, our study also uncovers temporal variability in timing bud break which is enhanced under fluctuating temperatures. This variability in bud break, associated with bet hedging, may enhance survival under unpredictable conditions. Buds frequently experience unpredictable cold snaps in early spring after they reinitiate bud break[14,16,53]. Such cold snaps can cause fatal irreversible damage to the growing leaf primordia[16,54,55]. Variation in bud break timing, spreads bud break within a population, increasing the chances that some individuals escape damage as is the case with bet hedging. Interestingly, the variability in bud break is strongly reduced in *abi1-1* mutant buds. While bet-hedging is a well-studied universal response, the underlying mechanism in plants is not well understood[42,44]. Recently, variability in hypocotyl elongation and bolting in Arabidopsis and barley suggested that bet hedging is linked with ELF3, a prion-domain like protein mediating temperature responses[56]. Interestingly in the *abi1-1* mutant, the regulation of PD opening is uncoupled from temperature regulation and therefore it is tempting to speculate a possible role for PD dynamics contributing to

variability in bud break, a key adaptive response. While bet-hedging in populations can be anchored at the genomic level (via sequence variation), we show such variation can occur in genetically identical individuals and may involve regulation of cell–cell communication.

Robust regulation of dormancy release has a profound effect on the survival of trees[57] and chilling fulfillment controlling dormancy release has been studied for many years[31,32]. However, the mechanisms enabling chilling fulfillment for dormancy release under fluctuating temperatures have remained unknown. We identify the dynamic response of the PD network, as a key integrator of variable temperature processing. Inhibition of cell-cell communication delays chilling fulfillment under long warm spikes, revealing the mechanistic basis for variable temperature processing and robust temporal control of dormancy release.

Importantly, PD play a key role in the spatiotemporal control of diverse developmental transitions by controlling the cell-to-cell movement of regulators e.g., proteins, small RNAs and hormones (e.g., GA and brassinosteroids)[58–63]. Thus mechanistic framework revealed here could be extended also to understand the regulation of developmental processes controlled by other variable environmental information. Importantly, our studies showing how longer warmer spikes negatively influence dormancy release may also, provide an explanation at a molecular level for the delayed or inadequate dormancy release caused by warmer winters[64–66] as a consequence of climate change. Such delays are causing a decline in the productivity of fruit trees[67], which rely on very similar regulatory processes as the model hybrid aspen. Thus, our findings are of broader significance for understanding of adaptation of plants and their responses to future climate scenarios.

## Methods
### Plant material and growth conditions
Wild-type (WT) hybrid aspen (*Populus tremula × tremuloides*, clone T89) and transgenic plants were initially grown on half-strength Murashige–Skoog medium (Duchefa) under sterile conditions for four weeks. Following this, they were transferred to soil and grown in a

greenhouse with a controlled environment (18-h light/6-hour dark cycle at 22 °C during the day and 18 °C at night) for five weeks with fertilization. Subsequently, the plants were subjected to short-day (SD) conditions (8-h light/16-h dark cycle at 20 °C/15 °C) for 11 weeks to induce growth cessation and dormancy[21]. After 11 weeks under SD conditions, the plants were exposed to either constant cold (24 h at 4 °C) for 4 weeks or variable cold (20 h at 4 °C/4 h at 20 °C) (Fig. 1a). To reflect natural conditions, warm spikes were introduced during mid-day, when warm temperatures are typically encountered. For quantification of the bud break phenotype (Figs. 1c, 2d and Supplementary Fig. 11), all transgenic lines and the wild-type (WT) control were grown and analysed at the same time. For bet-hedging experiment, variable cold was 20 h at 4 °C and 2 h at 20 °C (Fig. 4a). The variable cold treatment was extended to ensure that dormant buds in variable cold received an equivalent number of cold hours as buds exposed to constant cold. Following a cold or variable cold, plants were transferred to long-day (LD) (18 h light/6 h dark) and warm-temperature conditions (20 °C) (LD/WT), and bud break was assayed by bud swelling and the emergence of green leaves. For gene expression analysis, apices of WT plants were sampled at different stages: after 11 weeks of SD treatment (11WSD), and after two weeks (2WC) and four weeks (4WC) of exposure to the constant or alternating cold/warm conditions. As indicated above, for variable cold, bud samples were collected so that buds had received an equivalent cold hours at 2 or 4 weeks as constant cold-treated buds at these time points. Each collected sample was immediately frozen in liquid nitrogen and stored at −80 °C for future analysis. Photographs of the apices were captured using a Canon EOS digital camera to document the bud break process.

## RNA isolation and qRT-PCR analysis
Total RNA was extracted from plant shoot apices using the Spectrum™ Plant Total RNA Kit (Sigma-Aldrich). RNA (10 µg) was treated with RNase-free DNaseI (Life Technologies, Ambion) to remove contaminating DNA. Subsequently, 1 µg of this RNA was used for cDNA synthesis using the iScript cDNA Synthesis Kit (BioRad). Ubiquitin was used as the reference gene in all experiments[24]. Quantitative reverse transcription PCR (qRT-PCR) was conducted using a LightCycler 480 SYBR Green I Master mix and a LightCycler 480 II instrument (both from Roche). The relative expression levels of target genes were calculated using the Δ-cq method, as previously described[68]. Details of the primer sequences used for qPCR are provided in Table S1.

## Generation of transgenic lines
To generate *abi1-1/ft1* transgenic lines, *FT1-pHSE401* (FT1-Crispr)[24] construct was transformed in *abi1-1* background via *Agrobacterium*-mediated transformation[21]. Generation of *GA20oxOE* transgenic line has been described previously (*36*). To generate inducible *XVE::FT1-GFP-HA* line, *FT1-GFP* fragment[69] was amplified using the primers listed in Table S1. The amplified fragment was cloned into pENTR/D-TOPO (Invitrogen) and subsequently transferred into the plant transformation vector pDMC7. Hybrid aspen transformation was performed as described previously[21].

## Estradiol treatment
Estradiol treatment was performed by gently removing the bud scales of plants that had undergone growth cessation and dormancy establishment. The exposed buds were then dipped in either 10 µM estradiol or DMSO solution. To prevent wilting, the plants were covered with a transparent bag. The plants remained in short-day conditions and were treated with estradiol every third day. For gene expression and PD analysis, the treated bud samples were collected after 2 weeks. For gene expression analysis, samples after collection were immediately frozen in liquid nitrogen. For bud break analysis, swelling and emergence of green leaves were monitored.

## Grafting experiment
FT1ox and *ft1* Crispr plants were grown in chambers under short photoperiod conditions (SD) (8 h, 20 °C light/16 h, 15 °C dark cycles, 80% relative humidity) for 11 weeks and transferred to cold for 5 weeks. Scions of the *ft1* plants were then grafted onto root stocks of *ft1 Crispr* and FT1-overexpressing plants, as described previously[21]. Grafted plants were kept under LD and monitored for bud break.

## Transmission electron microscopy (TEM) sample preparation, imaging, and PD mapping
Samples for TEM were prepared as described earlier[21]. Buds from 11 weeks of short days and 4 weeks of constant and variable cold temperature after removing bud scales and outer leaf primordia were fixed at room temperature (RT) in a slowly agitating shaker overnight in a solution containing 2% paraformaldehyde, 2% glutaraldehyde in 0.05 M cacodylate buffer, and 10 mg/ml tannic acid. Samples were then rinsed in cacodylate buffer (0.05 M) and post-fixed in 1% osmium tetroxide (in water) for 1 h at RT in darkness. Following post-fixation, samples were dehydrated through a graded ethanol series (30–100%) and propylene oxide (100%), then embedded in Spurr resin (S031/D) in casting molds and polymerized at 70 °C for 24 h.

The fixed buds embedded in Spurr blocks were longitudinally sectioned into 90 nm thick slices using an EM UC7 ultramicrotome (Leica) and placed onto 100 mesh copper grids. Observations were carried out using a FEI TECNAI Spirit 120 kV electron microscope. To acquire data for PD mapping, high-resolution individual TEM images (at a magnification of 4400x) were captured and stitched together using the ImageJ software to reconstruct the entire meristem surface (Supplementary Fig. 14). PD were identified and counted manually in approximately 50 cells. The Inkscape software was then used to create a color map based on the acquired data, reflecting the number of PD at cell-to-cell interfaces (Supplementary Fig. 14).

## Callose immunostaining
The fixed buds embedded in Spurr blocks were longitudinally sectioned using an EM UC7 ultramicrotome (Leica). For immunofluorescence, 200 nm thick slices were placed onto poly-lysine-coated slides. For immunogold, 90 nm thick slices were placed onto 100 mesh copper grids. Both techniques followed the same steps up to the primary antibody incubation, with the difference starting from the secondary antibody step onward.

Right after sectioning, immunolabelling was performed using a modified version of a previously described protocol[70]. All solutions were prepared in PHEM buffer, comprising 60 mM PIPES, 25 mM HEPES, 10 mM EGTA, and 4 mM MgSO4·7H2O. To prevent nonspecific binding, the samples were subjected to a blocking step using 5% neutral donkey serum (NDS) before incubation with the primary antibody. The callose antibody (obtained from Australia Biosupplies) was appropriately diluted to 1/20 in PHEM buffer supplemented with 5% (v/v) NDS and incubated with the samples at room temperature for 2 h.

For immunofluorescence, after washing with PHEM buffer (5 times for 5 min each), the secondary anti-mouse antibody, coupled to Alexa Fluor 555, was diluted to 1/100 in PHEM buffer containing 5% (v/v) NDS. This antibody solution was then applied to the samples and incubated for 2 h. Following another series of washes in PHEM buffer (5 times for 5 min each), 1 mg/ml of calcofluor was applied for 10 min to stain the cell walls. Finally, the slides were rinsed with Milli-Q water.

For immunogold labeling, after washing with PHEM buffer (5 times for 5 min each), the secondary anti-mouse antibody, coupled to 10 nm EM gold particles, was diluted to 1/30 in PHEM buffer containing 5% (v/v) NDS. This antibody solution was then applied to the samples and incubated for 2 h. Following another series of washes in PHEM buffer (5 times for 5 min each), the samples were finally rinsed with Milli-Q water.

## TEM imaging and callose quantification

Immunogold-labeled sections were imaged using a FEI TECNAI Spirit 120 kV electron microscope. High-resolution individual TEM images (at a magnification of 42,000x) were captured to quantify callose deposition by counting the number of gold particles. For normalization, a rectangular region of $0.025\,\mu m^2$ was delineated around the plasmodesmata. This quantification was performed across different genotypes and experimental conditions.

## Confocal microscopy

All confocal imaging experiments were conducted on a Zeiss LSM 880 microscope, operated with ZEN Black 2011 software. The slides, obtained from immunofluorescence, were mounted with a thin layer of Milli-Q water and a coverslip, and images were captured using a Plan-Apochromat 40x oil objective. Alexa Fluor 555 fluorescence was detected using an Ex 561 nm and Em 561–615 nm filter, while calcofluor fluorescence was detected using an Ex 405 nm and Em 411–476 nm filter.

## Data extraction, analysis and display

Confocal microscopy stacks with two channels, one labeling cell walls and the other staining PD with callose, were used (Supplementary Fig. 15A). A custom ImageJ toolset was employed for the automated segmentation of cells, walls, and PDs while preserving relational information (Supplementary Fig. 15B–C). The toolset includes tools for CZI to Zip conversion, single file analysis, multiple file analysis, and data compression.

The analysis involved several key steps. The original wall image was pre-processed to enhance and skeletonize the walls. Tri-junction points (points where three walls converge) were identified and used to segment the skeleton into individual wall segments. Cells were then identified based on the areas enclosed by walls. For PD, the original image was pre-processed, and local maxima were detected. The toolset then extracted relational data: Each PD was associated with a wall, each wall segment with two adjacent cells, and each junction point with three connected walls.

The toolset outputs composite images with multiple channels allowing visual inspection of detections, CSV files containing data on cells, junction points, PDs, and walls, a comprehensive JSON file containing all extracted data, analysis parameters, and image metadata, and PNG image files of each channel. The toolset exposes parameters for preprocessing, detection, and parent search tolerances, allowing users to adjust the analysis according to specific needs.

The toolset, example datasets, and example output are available from the following GitHub: https://github.com/fabricecordelieres/IJ-Toolset_PD-analysis/tree/main.

Further analysis and data visualization are made possible using a Python script. Briefly, the script generates several visual outputs to present PD density at several granularities (per wall, per length of wall, per cell). It also outputs numerical data in the form of table data to ease further statistical analysis. For each cell-cell interface, the wall segment length is exported, along with the number of PD and their total intensity, allowing to extract their density. The Python code, example datasets, and example output are available from the following repository: https://gitub.u-bordeaux.fr/gmaucort/plasmodesmata_import_display_report.

## Reporting summary

Further information on research design is available in the Nature Portfolio Reporting Summary linked to this article.

# Data availability

All data supporting the findings of this study are available within the paper and its Supplementary Information files. All materials generated in this study are available from the lead contact upon request. Source data are provided with this paper.

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

## Acknowledgements

We would like to thank Kristoffer Jonsson for helping with model figure and Olivier Hamant and Christian Hardtke for reading and commenting on the article. All light and electron imaging were done at the Bordeaux Imaging Center, member of the national infrastructure France-BioImaging supported by the French National Research Agency (ANR-10-INBS-04). This work was supported by Human Frontier Research program, Project: RGP0002/2020 (S.W., G.B., E.M.B., R.B.), European Union's Horizon 2020 research and innovation program under the Marie Sklodowska-Curie grant agreement No 890883 (DECORE: SEP-210709191) (S.K.P., R.B.), Wenner-Gren fellowship UPD2019-0203 (S.K.P., R.B.), Knut and Alice Wallenberg Foundation grant 2023.0209 (R.B.), Vetenskapsrådet 2020-03522 (R.B.), The French government in the framework of the IdEX Bordeaux University "Investments for the Future" program / GPR Bordeaux Plant Sciences (EMB), The European Research Council (ERC) under the European Union's Horizon 2020 research and innovation program, Project: 772103-BRIDGING (EMB), BBSRC grant no. BB/S002804/1 (G.V.D., G.B.) and Leverhulme Trust Grant RPG-2019-267 (G.V.D., G.B.).

## Author contributions

R.B., E.M.B., G.W.B. and S.W. conceived and supervised the study. S.K.P., T.S.M., A.N., B.A., A.A., P.M., G.M., F.P.C., L.B., G.V.D., H.D. and S.K. developed the methodology. S.K.P., T.S.M., A.N., B.A., P.M., G.M., F.P.C. and L.B. performed validation and data visualization. R.B., E.M.B., G.W.B., S.W., S.K.P., T.S.M. and A.N. carried out the investigation. R.B., E.M.B., G.W.B., S.W., S.K.P., T.S.M. and A.N. wrote the manuscript. All authors reviewed and approved its final version.

## Funding

## Competing interests

The authors declare no competing interests.
