## [Transparent Peer Review file · Nature Communications]

Variable Temperature Processing by Plasmodesmata Regulates robust Bud Dormancy release

Corresponding Author: Professor Rishikesh Bhalerao

Version 0:

Reviewer comments:

Reviewer #1

(Remarks to the Author)

This research group recently published a lot of important findings about tree bud dormancy using genetic approaches. Especially, they reported that ABA has central role for bud dormancy establishment and proposed the dense TEM signal in plasmodesmata (they called plasmodesmata sphincter, Tylewicz et al., 2018, Science) is found only in normal WT but not in *abi1-1* mutant, suggesting it is critical morphological marker of dormancy establishment. They also reported that low-temperature-induced dormancy release (later stage) is mediated by LIM1, FT1 and GA pathways (Pandey et al., 2024, EMBO J.). These published studies are consistent with important pioneer findings published by van der Schoot group [Rinne et al. (2001), Rinne et al. (2011)], which have facilitated recent bud dormancy study.

This study demonstrated that warm spikes during dormancy release period suppress bud break through the thermal responsiveness of plasmodesmata. Moreover, GA pathways are also involved in this response. Quantification of callose levels in plasmodesmata using callose-specific antibody and genetic approach using transgenic plants provided solid evidence to support some parts of their thoughts.

My first concern is the appropriateness of the use of the word “robust bud dormancy release” in the title. As previously reported, dormancy sphincter removal should be the right word of “dormancy release”, to my understanding. Therefore, the process from plasmodesmata closure to opening may represent “later stage of dormancy release” or “the stage towards bud break”. The topic in this study is not about dormancy sphincter removal but PD opening, which corresponds to “late dormancy release towards bud break”. Therefore, I suggest the use of “bud break” rather than “dormancy release” for the title. Fig5 (a) model does not represent “dormancy release” but “bud break transition”.

My second concern is the link between bet hedging (heterogeneity in bud break) and plasmodesmata. Although the authors proposed there is close link between them in the abstract and Fig. 4, the authors did not investigate callose level in buds under variable cold (2h-W). They also did not investigate callose level in *abi1-1* mutant at all. Under this circumstance, the relationship between plasmodesmata and bud break heterogeneity or bet-hedging has not been well-validated in this study. Fig4 title is not appropriate. ABA controls PD sphincter (Tylewicz et al., 2018) but ABA also important for callose level in PD? Is this validated?

There are other comments

1. While FT1 and GA20OX show similar expression patterns under the 11WSD, Cold, and Variable Cold treatments, this observation alone may not be sufficient to clarify how FT1 regulates PD opening independently of the LIM1 and SVP pathways.

Additional mechanistic evidence would help substantiate the proposed model.

2. In Fig. 3. Including *ft1* knockout comparisons under both Cold and Variable Cold conditions would substantially strengthen the conclusions. This would clarify whether FT1 is essential for temperature-induced dormancy release and whether its reduced expression under Variable Cold is a primary cause of dormancy failure. Without this analysis, the proposed role of FT1 remains speculative.

3. The authors present an interesting and valuable set of phenotypic data for *abi1-1*, (*ft1*, should be included) and the *abi1/ft1* double mutant under various environmental treatments. However, the underlying regulatory mechanisms remain unclear. To better understand the genetic interactions and clarify whether ABI1 and FT1 act in the same or parallel pathways, it would be helpful to analyze the expression levels of key genes such as SVL, LIM1, FT1, and GA20ox in these mutant backgrounds. Specifically, comparing gene expressions across 11WSD, constant cold, and variable cold conditions could provide more

direct evidence of the regulatory roles and relationships among these genes. This would significantly strengthen the mechanistic interpretation of the observed phenotypes under variable cold.

Fig. 1b, Fig. S7- Could the authors clarify why the WT plant in Fig. 1b (11WSD) appears to retain its leaves, while the WT plant shown in Fig. S7 (10WSD) seems to have entered dormancy and shed its leaves? This apparent inconsistency raises some confusion—was there a mistake in photo selection or labeling? Or is there another explanation?

Reviewer #2

(Remarks to the Author)

Pandey et al. address how poplar buds are released from dormancy by environmental signals, thus adding to an existing body of work on this highly important physiological response of plants and the end of the winter season. The authors describe and investigate a previously insufficiently characterised mechanism, whereby poplar plants suppress bud break in response to fluctuating outside temperature. Understanding plant responses to such "real life" scenarios and variable environments is highly relevant, not only for our fundamental understanding of plant development, but also to explore and possibly predict plant responses to changing current and future climate scenarios. The authors investigate the expression and functional role of key regulators of bud break in poplar, including transcription factors, hormones, and mobile factors that are known as floral inducers, such as FT1. The manuscript reports on a key role for FT1 in triggering dormancy release, mediated by opening of plasmodesmata by reducing callose levels, and how short warm periods that interrupt cold phases can basically erase the effect of extended cold, resulting in suppression of bud break. The authors propose that FT1 is a key factor in this response, and that the underlying mechanism supports bet hedging, i.e. a differential response of buds to the same environmental input in order to increase reproductive success.

The experiments are well presented and described. I very much enjoyed the high quality of the analysis of callose deposition at PDs, based on antibody detection using fluorescence labelling for entire meristems, and detailed surveys via electron microscopy with gold labelled antibodies! Overall, a well written manuscript which addresses dormancy and the role of plasmodesmata in bud break from a new angle, which was a pleasure to read.

Nevertheless, there are a few points that the authors should consider and address to further strengthen the manuscript, which I have detailed below.

1) A key point the authors make in the manuscript is that FT1 expression is sufficient and can substitute for cold in inducing PD opening and dormancy release. This statement is based, among other experiments, on an inducible FT1 transgenic line, which can overcome environmental regulation and induce bud break and PD opening (judged by reduction in callose). However, the authors have analysed changes in callose accumulation at PDs several weeks after FT1 induction, so FT1 might be a rather indirect regulator of PD callose. This problem (correlation, causality) could be resolved by assaying PD callose levels shortly after FT1 induction, and before bud break. In that context, I would like to quote a sentence from the manuscript, which highlights this more: "Altogether, these results indicate that warm spikes act via FT1 to suppress downregulation of callose which then prevents dormancy release". It would be equally possible to argue that warm spikes act via FT1 (and possible LIM1) to prevent dormancy release, which would normally result on PD opening and callose reduction.

2) To determine whether defects in callose deposition/degradation could affect bud dormancy break the authors used the *abi1* mutant, which was previously reported to break dormancy even in cold condition, because plasmodesmata are always open in this mutant background. They showed here that the mutant displayed dormancy break also in the variable cold condition, i.e. with warm spikes interrupting the cold period.; also, they are insensitive to the warm spikes, that inhibit dormancy break in wt condition. Determining the level of callose at PDs in the three conditions (11WSD, constant cold and variable cold) in the *abi1* background is important to evaluate whether a decrease in callose accumulation at the plasmodesmata is necessary and sufficient to trigger break dormancy.

3) One feature to assess bud dormancy is the presence of lipid droplets. As a matter of fact, the authors nicely observed lipid droplets in the 11WSD treatment. The information whether also buds from variable cold condition showed lipid droplets is missing (lipid droplets of what??)

4) A reported pathway to trigger dormancy release relies upon the simultaneous downregulation and upregulation of SVL and LIM1, respectively. The authors therefore assayed expression of SVL and LIM1 in entire buds by qRT-PCR in the three different conditions. They point out that under both continuous cold and warm spiked conditions, SVL is still repressed, and LIM1 is still induced. FigS4 shows, in my opinion, a strong difference in the response of LIM1 under the two key conditions. Is that statistically different? Do all cells in a bud behave in the same manner, i.e., could there be regional differences in expression levels? A possible interpretation of the data, which differs from the authors point of view, could be that LIM1 is not sufficiently induced in the warm-spiked situation, and therefore dormancy is not broken.

5) FT1 has been previously reported to be a cold induced promoter of dormancy release. To determine whether inhibition of dormancy release in variable cold condition was due to a failure in triggering FT1 expression, the authors performed qRT-PCR on entire buds, testing FT1 expression in the three conditions. They observed a decrease in FT1 expression in variable cold condition compared to the constant cold, suggesting that the failed dormancy release in variable cold condition was caused by reduced levels of FT1 expression in the buds. Statistical analysis would strengthen this dataset; however, given that LIM1 expression levels were also affected in variable cold conditions (compared to constant cold), focussing only on FT1 now seems to ignore other possible pathways and scenarios.

6) The authors showed here that the *ft1* mutant buds do not show a decrease in callose (associated to the number of gold particles) in cold condition. Different callose levels could also be caused by a different number of PDs. Analyzing the number of plasmodesmata in *ft1* mutant would be important to determine how FT1 act in dormancy release. Furthermore, the data presented here are in contrast to data presented by André et al., 2022, which uncouple FT1 function and callose presence at PDs in poplar (DOI: 10.1016/j.cub.2022.05.023). This should be discussed.

7) The authors showed that the double mutant *abi1-1/ft1* presented bud break even in variable cold condition. They

concluded that warm spikes act via FT1 to suppress downregulation of callose to prevent bud break. I think that their conclusions are not supported by the data and additional observations are needed. Murata et al., (2025) very recently reported that cold triggers ABA signaling that in turn induces callose synthesis by upregulation of CALLOSE SYNTHASE genes, restricting FT translocation to the SAM in *Arabidopsis thaliana*. The authors here suggested a direct activity of FT1 on degradation/deposition of callose, modulating dormancy break. Analysis of FT1 translocation in *abi1-1* mutant would be important to address whether movement of FT1 in the bud is important for dormancy break or FT1 solely act in modulating callose distribution at PDs. Do expression patterns of FT1, and local decrease of callose overlap, or does this occur in different, separate cells? I.e. is FT1 transport required for the observed effects?

8) Based on their results, the authors concluded that FT1 (together with GA) trigger dormancy break but their relationship is not further investigated. Enhanced expression of FT1 could be the effect and a consequence of callose degradation/reduced callose deposition at PDs and not the trigger of the response. Analysis of FT1 expression in the GA overexpression lines and analysis of dormancy release in GA20oxoOE *ft1* double mutant would clarify this aspect.

9) The authors here reported that level of expression of FT1 are lower in variable cold condition compared to cold condition; considering that the *ft1* mutant presented defects in dormancy break even in cold condition they concluded that warm spikes inhibit bud break. FT1 expression was quantified by qRT-PCR on the entire bud. Analysis of the pattern of expression of FT1 would determine whether the accumulation of callose is inhibited in the same cells where FT1 expression is triggered or if FT1 acts non-cell autonomously to determine dormancy break.

10) LIM1 regulation is one of the key responses to cold, and upregulation is reduced in variable cold conditions (Fig S4). could you provide some statistical analysis of these data? Are these qRT-PCR data? How would LIM1 mutants respond to variable cold, i.e., is LIM1 here the key factor that mediates the difference in response?

11) The suggested bet-hedging strategy is certainly intriguing, but not really explored sufficiently in this manuscript. As stated above (point 9), I am missing the spatial component in the analysis of bud break. Are the breaking/non-breaking buds evenly distributed at the plant? How is the response of a bud, consisting of many hundred of cells, overall integrated? I am tempted to suggest to take this part of the story out of the manuscript, and explore this in-depth in a separate manuscript. It appears here like a very interesting, but insufficiently supported add-on.

Reviewer #3

(Remarks to the Author)

Reviewer #4

(Remarks to the Author)

Reviewer #5

(Remarks to the Author)

This study provides important mechanistic insight into how plants integrate variable temperature signals to regulate bud dormancy release. The authors demonstrate that plasmodesmata-mediated cell–cell communication acts as a key integrator of fluctuating thermal cues, particularly through the detection and interpretation of intermittent warm spikes. They show that long warm periods suppress dormancy release by preventing PD reopening via repression of FT1 and GA20-oxidase expression. In contrast, shorter warm interruptions allow cold integration and promote dormancy release. These findings reveal a previously uncharacterized role for FT1 in callose downregulation and PD opening. The study also links temperature-responsive PD dynamics to phenotypic variability in bud break, suggesting a cellular basis for bet-hedging under unpredictable spring conditions. Overall, this work presents a compelling and original framework for understanding how plants maintain robust developmental timing amid environmental noise, with broader relevance to plant adaptation under climate change.

The manuscript is well written, and the data presented are convincing, novel, and well suited for publication in Nature Communications. Below are a few comments and questions for the authors' consideration:

Comments:

Figure 1a and Figure 4a

The schematic suggests that warm spikes were applied at different times on consecutive days (e.g., from ZT20–ZT24 on the first day, and from ZT40–ZT44 on the second). Is this interpretation correct? If so, could the authors clarify the rationale for this design?

Figure 2, Figures S1 and S3 and discussion

The observation of stronger callose deposition at PDs in the SAM compared to adjacent tissues raises an interesting possibility of cold-induced, cell–cell communication regulation. Do FT1 or GA biosynthesis genes exhibit SAM-specific expression patterns under these conditions?

Could this reflect SAM-specific expression of β -1,3-glucanases involved in PD opening?

Line 119

"In agreement with prior studies, callose levels decreased in dormant buds exposed constant cold (Fig. 2a, Fig. S1a, Fig. S2a)."

Could the authors clarify how the measured callose levels were specifically linked to PD openness? This connection is not immediately evident from the provided images.

Line 238

The data show that constitutive overexpression of GA20ox leads to bud break in nearly 50% of plants under variable cold conditions (Fig. S6). However, this phenotype does not fully phenocopy the effects of FT1 induction. Could the authors elaborate on potential reasons for this divergence? Does this suggest that FT1 may regulate additional GA-independent pathways required for PD opening and dormancy release?

Line 292-297

if cold integration mechanism that enables robust sensing of autumn to late autumn/winter transition, as, typically, there is a shift towards progressively shorter periods of warm spikes relative to cold. A lack of response to cold when the warm spikes last longer can therefore ensure that cold registration is not initiated and chilling threshold is not achieved prematurely, preventing precocious dormancy release..... (Fig.5)

Is callose deposition at PDs, and the resulting closure of cell-cell communication, only triggered when warm spikes become short enough? If so, at what point in this model does PD dynamics shift from closure to opening?"

Version 1:

Reviewer comments:

Reviewer #1

(Remarks to the Author)

I checked the revised version and found that the authors provided a lot of new data and revised the manuscript properly, which addressed all my concerns. Regarding the title, I understand the author's explanation and agree to retain the original title for this revised manuscript.

Reviewer #2

(Remarks to the Author)

Thank you for the new version of the manuscript. I feel that the authors have addressed all concerns that were raised by me and the other reviewers.

Reviewer #3

(Remarks to the Author)

Reviewer #4

(Remarks to the Author)

Reviewer #5

(Remarks to the Author)

Dear Authors,

The revised version shows significant improvement, including clearer explanations and the addition of new supporting data that strengthen the overall work.

In my opinion, the manuscript now meets the journal's standards and should be accepted for publication.

Dear Editor,

Please find enclosed our detailed response to reviewers. We have addressed the reviewers' comments point by point experimentally and adding new results to the revised manuscript as well as via additional clarifications and revised the manuscript accordingly. We hope our response with the revisions will be satisfactory for acceptance of our manuscript for publication.

Sincerely,

Rishikesh Bhalerao

Response to reviewers:

Reviewer #1 (Remarks to the Author):

This research group recently published a lot of important findings about tree bud dormancy using genetic approaches. Especially, they reported that ABA has central role for bud dormancy establishment and proposed the dense TEM signal in plasmodesmata (they called plasmodesmata sphincter, Tylewicz et al., 2018, Science) is found only in normal WT but not in *abi1-1* mutant, suggesting it is critical morphological marker of dormancy establishment. They also reported that low-temperature-induced dormancy release (later stage) is mediated by LIM1, FT1 and GA pathways (Pandey et al., 2024, EMBO J.). These published studies are consistent with important pioneer findings published by van der Schoot group [Rinne et al. (2001), Rinne et al. (2011)], which have facilitated recent bud dormancy study. This study demonstrated that warm spikes during dormancy release period suppress bud break through the thermal responsiveness of plasmodesmata. Moreover, GA pathways are also involved in this response. Quantification of callose levels in plasmodesmata using callose-specific antibody and genetic approach using transgenic plants provided solid evidence to support some parts of their thoughts.

My first concern is the appropriateness of the use of the word “robust bud dormancy release” in the title. As previously reported, dormancy sphincter removal should be the right word of “dormancy release”, to my understanding. Therefore, the process from plasmodesmata closure to opening may represent “later stage of dormancy release” or “the stage towards bud break”. The topic in this study is not about dormancy sphincter removal but PD opening, which corresponds to “late dormancy release towards bud break”. Therefore, I suggest the use of “bud break” rather than “dormancy release” for the title. Fig5 (a) model does not represent “dormancy release” but “bud break transition”.

Response:

We thank the reviewer for raising an important point. We are happy to revise the title to include the reviewer's point of view to: **Variable Temperature Processing by Plasmodesmata Regulates robust Dormancy release preceding bud break** if necessary.

However we would like to include a clarification regarding the relationship between analysis of dormancy sphincters and callose related to PD and dormancy. In buds, bud break can only proceed

if dormancy is released. Crucially, in context of dormancy regulation, PD closure is essential for dormancy establishment whereas their opening is essential for dormancy release as shown by prior studies by Rinne et al and others. Accordingly, Rinne and others (including us) have interchangeably used either callose quantification (using anti-callose antibodies) or “dormancy sphincters” (that are visualized by TEM as electron dense structures) as a marker for PD closure/opening associated with dormancy establishment or its release. Ours as well as results of others show that analysis of dormancy sphincters or callose quantification basically give same results. For example, in the wild type at 11 WSD both callose levels and dormancy sphincters are high whereas after cold treatment at both dormancy sphincters and callose levels are low. Similarly, in *abi1-1* we have previously shown that dormancy sphincters are very low at 11 WSD and we show here (see below Fig. 2) that callose levels are also very low, consistent with the open PD phenotype. Thus, the status of closure or openness of PD associated with dormancy regulation can be measured either with loss of dormancy sphincters or alternatively by callose measurements. Here, we chose quantification of callose levels (although they are far more challenging to do) than looking at dormancy sphincters since callose quantification provides a better cellular resolution for PD opening (as a prerequisite for dormancy release). Based on this clarification we would appreciate if the reviewer would consider to agree with us to retain original title. If not we can change the title as indicated.

My second concern is the link between bet hedging (heterogeneity in bud break) and plasmodesmata. Although the authors proposed there is close link between them in the abstract and Fig. 4, the authors did not investigate callose level in buds under variable cold (2h-W). They also did not investigate callose level in *abi1-1* mutant at all. Under this circumstance, the relationship between plasmodesmata and bud break heterogeneity or bet-hedging has not been well-validated in this study. Fig 4 title is not appropriate. ABA controls PD sphincter (Tylewicz et al., 2018) but ABA also important for callose level in PD? Is this validated?

Response:

We thank the reviewer, and to address this concern, we have now provided two additional evidences as requested. First, we measured callose levels under 2 hour warm spikes. Our data shows that in contrast with 4 hour spikes, 2 hour spikes are able to significantly reduce callose levels, and this reduction correlates with dormancy release (this data has been added as sup Fig S13).

As a response to reviewer’s suggestion, we have also changed the title of Fig. 4 to:

The bet-hedging response of buds is altered in *abi1-1* mutant

Fig. 1 Callose quantification at PD in the shoot apical meristem of aspen buds after 11 weeks of short days and in response to 4 weeks of constant or variable cold (cold exposure followed by 2 hours at 20 °C) temperatures. (A) Graph showing callose quantification per PD based on mean signal intensity (each data point represents an individual PD). Quantification was performed on six sample surfaces per condition, each with an analyzed area of 7515 μm^2 . Statistical analysis was conducted using one-way ANOVA ($p < 0.0001$), followed by Tukey’s test. Error bars represent the 95% confidence interval of the difference, and asterisks (****) denote a significant difference. (B) Callose immunofluorescence (purple) in semi-thin sections of aspen tissue visualized by confocal microscopy. Cell walls (grey) were stained with the fluorescent dye calcofluor. Callose accumulation (purple signal) was detected using an Alexa Fluor 555–conjugated secondary antibody. Scale bars: 10 μm .

Second, regarding the *abi1-1* mutant, we now present a new data (see below) showing that after 11 weeks under short-day conditions, callose levels are already significantly lower in the *abi1-1* mutant than in wild-type plants, consistent with our previous findings based on analysis of “dormancy sphincters” (Tylewicz et al., 2018). We hope this addresses reviewers’ concern. This data has also been added to revised manuscript (as supp Fig S4).

Fig. 2 Callose quantification at PD in the WT and *abi1-1* shoot apical meristem after 11 weeks of short days. (A) Graph displaying callose quantification per PD based on mean signal intensity, (data points represent individual PD). Quantification was based on six sample surfaces per condition, with analyzed surface areas of 4063 μm^2 . Statistical

analysis was performed using Mann-Whitney test ($p < 0.0001$). Error bars represent the 95% confidence interval of the difference, and asterisks (****) denote a significant difference. (B) Callose immunofluorescence (purple) in semi-thin sections of aspen tissue visualized by confocal microscopy. Cell walls are shown in grey, stained with the fluorescent dye calcofluor. The purple signal indicates callose accumulation, detected using Alexa Fluor 555-conjugated secondary antibody. Scale bars: 5 μm .

There are other comments

1. While FT1 and GA20OX show similar expression patterns under the 11WSD, Cold, and Variable Cold treatments, this observation alone may not be sufficient to clarify how FT1 regulates PD opening independently of the LIM1 and SVP pathways. Additional mechanistic evidence would help substantiate the proposed model.

Response:

We have shown earlier that FT1 redundantly of LIM1 regulates GA pathway that is promoter of PD opening (Pandey et al., 2024). However to further address reviewers' comment, we have now added data from inducible FT1 line (XVE:FT1) to show that FT1 induction in dormant buds in SDs, without cold treatment is sufficient to induce GA20-oxidase and suppress the GA2-oxidase. Thus this data shows that FT1 induction positively regulates GA pathway, which itself has been shown earlier to open PD (Pandey et al., 2024). Additionally, we show that FT1 induction modulates the expression of several other genes that are associated with PD regulation which bolsters the link between FT1 regulation of PD. We think these new data added to the revised manuscript (as supp Fig. S9 and S10) addresses reviewers' comment and thus provide explanation for FT1 link with PD regulation that is independent of SVL or LIM1.

Fig. 3: FT1 induces the expression of growth regulator genes. (A-C) RT-PCR data showing relative levels of GA20OX and GA2ox2 transcripts in buds of FT1-inducible lines treated with Mock (DMSO) or Estradiol. Plants were shifted to SD conditions for bud formation, the buds were treated with estradiol in SD and samples were collected after 2 weeks of treatment. Data represents the mean (\pm SEM) of three or more biological replicates and are normalized to UBQ. Asterisks indicate significant differences (* $p < 0.05$, **** $p < 0.0001$) calculated using t-test.

Fig. 4: FT1 induces the expression of PD regulator genes. (A-C) RT-PCR data showing relative levels of β -1,3-GLUCANASE 1, PDL6-2, and MCTP7 transcripts in buds of FT1-inducible line (XVE::FT1-GFP) treated with Mock (DMSO) or Estradiol. Plants were shifted to SD conditions for bud formation, the buds were treated with estradiol in SD and samples were collected after 2 weeks of treatment. Data represents the mean (\pm SEM) of three or more biological replicates and are normalized to UBQ. Asterisks indicate significant differences (* $p < 0.05$, *** $p < 0.001$, **** $p < 0.0001$, ns-not significant) calculated using t-test.

2. In Fig. 3. Including *ft1* knockout comparisons under both Cold and Variable Cold conditions would substantially strengthen the conclusions. This would clarify whether FT1 is essential for temperature-induced dormancy release and whether its reduced expression under Variable Cold is a primary cause of dormancy failure. Without this analysis, the proposed role of FT1 remains speculative.

Response:

As suggested by the reviewer, we show that *ft1* mutant fails to undergo dormancy release even under constant cold conditions in contrast with the wild type (Pandey *et al.* 2024). However we also present data that shows that *ft1* mutant does not undergo dormancy release also under variable cold. This data has been included in Fig 3g and Fig S11 and we hope this addresses the reviewers' comments.

3. The authors present an interesting and valuable set of phenotypic data for *abi1-1*, (*ft1*, should be included) and the *abi1/ft1* double mutant under various environmental treatments. However, the underlying regulatory mechanisms remain unclear. To better understand the genetic interactions and clarify whether ABI1 and FT1 act in the same or parallel pathways, it would be helpful to analyze the expression levels of key genes such as SVL, LIM1, FT1, and GA20ox in these mutant backgrounds. Specifically, comparing gene expressions across 11WSD, constant cold, and variable cold conditions could provide more direct evidence of the regulatory roles and relationships among these genes. This would significantly strengthen the mechanistic interpretation of the observed phenotypes under variable cold.

Response:

We thank the reviewer for the suggestion. The relationships between various factors e. g SVL, LIM1, FT1 and GA and ABA pathways are already published or included in the current manuscript. Our data in the manuscript shows that warm spikes do not act via SVL and LIM1 as these are regulated more or less the same in variable cold as constant cold (Fig. S5). Moreover in

ft1 mutant, *SVL* or *LIM1* expression is not altered (see below) which is in agreement with these acting upstream of *FT1* and that *FT1* does not regulate their expression.

Fig.5 RT-PCR data showing relative levels of *LIM1* and *SVL* expression in the buds of WT and *ft1* mutant plants. Data represents the mean (\pm SEM) of three or more biological replicates and are normalized to *UBQ*.

Also we have already shown that ABA and GA act antagonistically in dormancy regulation earlier and that ABA suppresses GA pathway in dormancy (Singh et al 2018). Thus we feel that the expression analysis would not be helpful in addressing whether *abi1-1* and *FT1* act in same pathway.

However, for addressing whether *ABI1* and *FT1* act in same or parallel pathway, we refer to our genetic analysis (which we think addresses reviewer’s comment) in the manuscript. Our data shows that loss of *ft1* in *abi1-1* mutant background, does not alter the phenotype of *abi1-1* (which is insensitivity to warm spikes). This is consistent with ABA being required for PD closure and of *FT1* being subsequently required for PD opening in the response to cold. The analysis of *ft1* mutant which cannot open PD in response to cold and do not undergo dormancy release further supports the key role of *FT1* in PD opening. Thus when induction of *FT1* is suppressed by warm spikes, closed PD cannot be opened, with resultant failure of the wild type to undergo dormancy release. Putting together these observations our data shows that ABA and *FT1* converge, acting antagonistically on PD regulation (ABA closing and *FT1* opening PD) as summarized below. Consequently, when ABA pathway is suppressed (leading to open PD), *FT1* is no longer required and thus *abi1/ft1* mutant phenocopies *abi1* and this is further explained schematically in figure below. We hope that this addresses reviewers’ concern. However if the reviewer still wishes us to perform expression analysis we are happy to do that but this will not be possible within the time constraints of the revision process.

Fig. 6 In the wild type, ABA induces the closure of PDs in buds whereas exposure of buds to cold induces FT1 which then induces the opening of closed PD. In contrast to constant cold, presence of warm spikes suppresses FT1 induction by cold and prevents opening of PDs and dormancy release in the wild type. However, in *abil-1* mutant, when ABA response is blocked, PDs are open therefore *abil-1* mutant is insensitive to warm spikes that act to block opening of PDs by cold via FT1. Consequently, when PDs are open as in *abil-1*, FT1 is no longer required and therefore *abil-1/ft1* mutant is insensitive to warm spikes like *abil-1*.

Fig. 1b, Fig. S7- Could the authors clarify why the WT plant in Fig. 1b (11WSD) appears to retain its leaves, while the WT plant shown in Fig. S7 (10WSD) seems to have entered dormancy and shed its leaves? This apparent inconsistency raises some confusion - was there a mistake in photo selection or labeling? Or is there another explanation?

Response: Thanks for pointing this out. Apologies for confusion. The correct picture has been added.

Reviewer #2 (Remarks to the Author):

Pandey et al. address how poplar buds are released from dormancy by environmental signals, thus adding to an existing body of work on this highly important physiological response of plants and the end of the winter season. The authors describe and investigate a previously insufficiently characterised mechanism, whereby poplar plants suppress bud break in response to fluctuating outside temperature. Understanding plant responses to such "real life" scenarios and variable environments is highly relevant, not only for our fundamental understanding of plant development, but also to explore and possibly predict plant responses to changing current and future climate scenarios. The authors investigate the expression and functional role of key regulators of bud break in poplar, including transcription factors, hormones, and mobile factors that are known as floral inducers, such as FT1. The manuscript reports on a key role for FT1 in triggering dormancy release, mediated by opening of plasmodesmata by reducing callose levels, and how short warm periods that interrupt cold phases can basically erase the effect of extended cold, resulting in suppression of bud break. The authors propose that FT1 is a key factor in this response, and that the underlying mechanism supports bet hedging, i.e. a differential response of buds to the same environmental input in order to increase reproductive success. The experiments are well presented and described. I very much enjoyed the high quality of the analysis of callose deposition at PDs, based on antibody detection using fluorescence labelling for entire meristems, and detailed surveys via electron microscopy with gold labelled antibodies! Overall, a well written manuscript which addresses dormancy and the role of plasmodesmata in bud break from a new angle, which was a pleasure to read. Nevertheless, there are a few points that the authors should consider and address to further strengthen the manuscript, which I have detailed below.

1) A key point the authors make in the manuscript is that FT1 expression is sufficient and can substitute for cold in inducing PD opening and dormancy release. This statement is based, among other experiments, on an inducible FT1 transgenic line, which can overcome environmental regulation and induce bud break and PD opening (judged by reduction in callose). However, the

authors have analysed changes in callose accumulation at PDs several weeks after FT1 induction, so FT1 might be a rather indirect regulator of PD callose. This problem (correlation, causality) could be resolved by assaying PD callose levels shortly after FT1 induction, and before bud break. In that context, I would like to quote a sentence from the manuscript, which highlights this more: "Altogether, these results indicate that warm spikes act via FT1 to suppress downregulation of callose which then prevents dormancy release". It would be equally possible to argue that warm spikes act via FT1 (and possible LIM1) to prevent dormancy release, which would normally result on PD opening and callose reduction.

Response: Thank you for raising this point. When we initiated the experiment we first performed a test analysis of FT1 induction at various time points. Our data (see below) indicated that after 1 week estradiol treatment FT1 induction was negligible and significant induction of FT1 was only observed around 2 weeks. Based on this result, we chose the relevant timepoint when we could observe FT1 induction i.e. 2 weeks, and also performed analysis of callose at this time point. Thus there is good temporal correlation and no real delay of several weeks between timing of significant FT1 induction and corresponding downregulation of callose level. We are happy to include this data to clarify in revised manuscript.

Fig. 1: FT1 expression in inducible line after 1 and 2 week of estradiol treatment.

RT-PCR data showing relative levels of *FT1* transcript in the buds of FT1 inducible line (*XVE::FT1-GFP*) treated with mock (DMSO), or 20uM estradiol for 1 or 2 weeks. All values are means (\pm SEM) of three biological replicates and are normalized against *UBQ*. Plants were shifted to SD conditions (8 hrs light/16 hrs dark) to induce bud formation. Estradiol treatment was done in the SD condition and samples were collected for gene expression. Asterisks indicate significant differences (**** $p < 0.0001$, t-test analysis).

The link between FT1 and callose is also supported by the *ft1* mutant analysis which shows that in absence of FT1, callose levels fail to be reduced after cold treatment. Moreover our grafting assay (that are indicative of PD being closed or open) show that *ft1* mutant buds fail to reactivate when

grafted on FT1 overexpressors (added as supp Fig S6). This result also supports that *ft1* mutant presumably fails to open PD and thereby also bolsters the link between FT1 mediated callose downregulation and dormancy release.

Further supporting the link between FT1, callose and dormancy is our data which shows that FT1 regulates GA pathway, inducing GA20-ox and simultaneously suppresses GA2-ox both of which are connected with callose regulation. Last, FT1 regulates the expression of genes associated with PD regulation (see below). We hope our clarification and additional data have convinced the reviewer of the causal link between FT1 with PD regulation, and our conclusion on how warm spikes act via suppression of FT1 expression to prevent PD opening and dormancy release. We have now added new data in the revised manuscript as supp Fig. S9 and S10.

Fig. 2: FT1 induces the expression of growth regulator genes. (A-C) RT-PCR data showing relative levels of GA20OX and GA2ox2 transcripts in buds of FT1-inducible lines treated with Mock (DMSO) or Estradiol. Plants were shifted to SD conditions for bud formation, the buds were treated with estradiol in SD and samples were collected after 2 weeks of treatment. Data represents the mean (\pm SEM) of three or more biological replicates and are normalized to UBQ. Asterisks indicate significant differences (* $p < 0.05$) calculated using t-test.

Fig. 3: FT1 induces the expression of PD regulator genes. (A-E) RT-PCR data showing relative levels of CALS1, β -1,3-GLUCANASE 1, PDL6-2, SUS6, and MCTP7 transcripts in buds of FT1-inducible line(XVE::FT1-GFP) treated with Mock (DMSO) or Estradiol. Plants were shifted to SD conditions for bud formation, the buds were treated with estradiol in SD and samples were collected after 2 weeks of treatment. Data represents the mean (\pm SEM) of three or more biological replicates and are normalized to UBQ. Asterisks indicate significant differences (* $p < 0.05$, *** $p < 0.001$, **** $p < 0.0001$, ns-not significant) calculated using t-test.

2) To determine whether defects in callose deposition/degradation could affect bud dormancy break the authors used the *abi1* mutant, which was previously reported to break dormancy even in cold condition, because plasmodesmata are always open in this mutant background. They showed here that the mutant displayed dormancy break also in the variable cold condition, i.e. with warm spikes interrupting the cold period.; also, they are insensitive to the warm spikes, that inhibit dormancy break in wt condition. Determining the level of callose at PDs in the three conditions (11WSD, constant cold and variable cold) in the *abi1* background is important to evaluate whether a decrease in callose accumulation at the plasmodesmata is necessary and sufficient to trigger break dormancy.

Response: We thank the reviewer for their comment. We have now performed analysis of callose levels in *abi1-1* mutant buds at 11 weeks SDs. This new data (which is now included in revised manuscript Fig S4) (see below) shows that, already at 11 weeks under short-day conditions (11WSD), even before any cold treatment, callose levels in the *abi1-1* mutant are significantly lower than in wild-type plants. This is also consistent with our previous findings which show that PD are already open in *abi1-1* (Tylewicz et al., 2018). Also we have shown earlier, *abi1-1* buds grafted FT1 overexpressors are able to undergo bud break unlike wild type buds which further show that PD are open in *abi1-1* mutant.

Our data shows that warm spikes act by preventing PD opening by callose downregulation. However new data on callose analysis of *abi1-1* as well as previously published data (Tylewicz et al., 2018) show that callose levels are already low in *abi1-1* mutant and PD are open and thus regulation of PD opening is uncoupled from cold regulation in *abi1-1* mutant and this correlates with the insensitivity of *abi1-1* to warm spikes and its phenotype. The new and previous data on *abi1-1* thus strongly suggests that since PD are already open, variable cold which basically acts by

suppressing the opening of closed PD is not effective in *abi1-1* (as suppression of ABA response will prevent PD closure). We hope that this addresses reviewer's comment. However, if absolutely necessary, we can check callose levels in *abi1-1* at PD under cold and variable cold conditions.

Fig. 4: Callose quantification PD in the WT and *abi1-1* shoot apical meristem after 11 weeks of short days. (A) Graph displaying callose quantification per PD based on mean signal intensity, (data points represent individual PD). Quantification was based on six sample surfaces per condition, with analyzed surface areas of 4063 μm^2 . Statistical analysis was performed using Mann-Whitney test ($p < 0.0001$). Error bars represent the 95% confidence interval of the difference, and asterisks (****) denote a significant difference. (B) Callose immunofluorescence (purple) in semi-thin sections of aspen tissue visualized by confocal microscopy. Cell walls are shown in grey, stained with the fluorescent dye calcofluor. The purple signal indicates callose accumulation, detected using Alexa Fluor 555-conjugated secondary antibody. Scale bars: 5 μm .

3) One feature to assess bud dormancy is the presence of lipid droplets. As a matter of fact, the authors nicely observed lipid droplets in the 11WSD treatment. The information whether also buds from variable cold condition showed lipid droplets is missing (lipid droplets of what??)

Response:

As suggested by the reviewer, we did perform the analysis of lipid droplets (see below, Fig 5). In our hands, the lipid droplets were present not only in 11WSD but also in cold as well as variable cold throughout the bud i.e. in dormant as well as non-dormant buds.

Fig. 5: Transmission electron microscopy images showing lipid droplets (black circles) in shoot apical meristem cells of aspen buds after 11 weeks of short days (A, B), under cold (C, D), and variable cold (E, F) treatments. Scale bars represent 2 μm .

Since we find lipid droplets in 11 WSD, cold as well as variable cold, their analysis is not informative and we would therefore prefer not include them in the manuscript.

4) A reported pathway to trigger dormancy release relies upon the simultaneous downregulation and upregulation of SVL and LIM1, respectively. The authors therefore assayed expression of SVL and LIM1 in entire buds by qRT-PCR in the three different conditions. They point out that under both continuous cold and warm spiked conditions, SVL is still repressed, and LIM1 is still induced. FigS4 shows, in my opinion, a strong difference in the response of LIM1 under the two key conditions. Is that statistically different? Do all cells in a bud behave in the same manner, i.e., could there be regional differences in expression levels? A possible interpretation of the data, which differs from the authors point of view, could be that LIM1 is not sufficiently induced in the warm-spiked situation, and therefore dormancy is not broken.

Response: To clarify, LIM1 expression may appear to be reduced under warm spikes, but following reviewer's suggestion we performed statistical analysis and this reduction is statistically not significant. Therefore we feel that it is unlikely that the reduction of LIM1 contributes to warm spike effects on bud dormancy. As for regional differences, when bud dormancy is released and growth initiated, to our knowledge there is no regional difference in growth response as whole bud swells and hence we feel that investigating regional differences in LIM1 expression are likely to be non-informative and somewhat out of scope of this manuscript.

5) FT1 has been previously reported to be a cold induced promoter of dormancy release. To determine whether inhibition of dormancy release in variable cold condition was due to a failure in triggering FT1 expression, the authors performed qRT-PCR on entire buds, testing FT1 expression in the three conditions. They observed a decrease in FT1 expression in variable cold condition compared to the constant cold, suggesting that the failed dormancy release in variable cold condition was caused by reduced levels of FT1 expression in the buds. Statistical analysis would strengthen this dataset; however, given that LIM1 expression levels were also affected in variable cold conditions (compared to constant cold), focussing only on FT1 now seems to ignore other possible pathways and scenarios.

Response:

We agree, FT1 reduction is statistically reduced in warm spikes (indicated now) whereas LIM1 is not (see above). This is clearly indicated in the revised manuscript.

6) The authors showed here that the ft1 mutant buds do not show a decrease in callose (associated to the number of gold particles) in cold condition. Different callose levels could also be caused by a different number of PDs. Analyzing the number of plasmodesmata in ft1 mutant would be important to determine how FT1 act in dormancy release. Furthermore, the data presented here are in contrast to data presented by André et al., 2022, which uncouple FT1 function and callose presence at PDs in poplar (DOI: 10.1016/j.cub.2022.05.023). This should be discussed.

Response: We agree that the number of PD is important, especially when buds are not in a dormant state, as it can affect the trafficking of information between cells. We have now included additional data showing PD density in the *ft1* mutant under both 11WSD and 4WC conditions. Our analysis shows that there is no significant difference in PD density between the *ft1* mutant and wild-type plants under these conditions. This data is included in revised manuscript (Fig. S7) and discussed.

Fig. 6: PD density do not change during cold treatment in the *ft1* mutant buds. PD were detected by callose immunofluorescence in the shoot apical meristem after 11 weeks of short days and in response to 4 weeks of constant cold regimes. Each callose dot was extracted as being one PD. **(A)** The graph shows the density of PD per wall. Quantification was based on six meristem surfaces per condition, each with an area of 6635 μm^2 . Statistical analysis was performed using Mann-Whitney test ($p < 0.0001$). Error bars represent the 95% confidence interval of the difference. **(B)** Representative meristem, with color-coded cell-cell interfaces. The color legend bar indicates the relative density, normalized to the cell wall length.

Thanks for pointing out the Andre et al. paper. To clarify, in contrast with what we have done here, Andre et al did not measure PD associated callose levels in *ft1* mutant. However their as well as (our unpublished) grafting data clearly indicates that PD are not open i. e. *ft1* buds grafted on FT1 overexpressors fail to induce bud break in line with PD being closed when FT1 is inactivated. Thus clearly in *ft1* mutant PD do not open which would be in agreement with a failure to downregulate callose levels and we have now indicated this in results section as suggested.

7) The authors showed that the double mutant *abi1-1/ft1* presented bud break even in variable cold condition. They concluded that warm spikes act via FT1 to suppress downregulation of callose to prevent bud break. I think that their conclusions are not supported by the data and additional observations are needed. Murata et al., (2025) very recently reported that cold triggers ABA signaling that in turn induces callose synthesis by upregulation of CALLOSE SYNTHASE genes, restricting FT translocation to the SAM in *Arabidopsis thaliana*. The authors here suggested a direct activity of FT1 on degradation/deposition of callose, modulating dormancy break. Analysis of FT1 translocation in *abi1-1* mutant would be important to address whether movement of FT1 in the bud is important for dormancy break or FT1 solely act in modulating callose distribution at

PDs. Do expression patterns of FT1, and local decrease of callose overlap, or does this occur in different, separate cells? I.e. is FT1 transport required for the observed effects?

Response: Thank you for pointing out the very nice Murata et al paper. We have referred to their work and compared it with ours in revised discussion. The findings of Murata et al are completely in line with our previously published data showing that ABA is required for callose deposition whereas blocking ABA response suppresses callose accumulation and keeps PD open (Tylewicz et al 2018) as well as new data on callose analysis in *abi1-1* buds (Fig. S4). However where trees differ from Arabidopsis is that in tree buds, short days induce ABA and this induces callose deposition to close PD whereas in Arabidopsis it is cold inducing ABA to close PD.

Here we are looking into the process of opening of closed PD during dormancy release, a process that is mediated by cold in tree buds and all our data (genetic and callose analysis of *ft1* mutant, analysis of FT1 inducible line) suggests that FT1 is induced by cold and FT1 function is essential for opening of PD and since opening of PD is essential for dormancy release, FT1 links PD opening and dormancy release.

Overall, our data suggesting FT1 downregulation by warm spikes suppresses FT1 expression and this downregulation of FT1 results in suppression of callose downregulation and failure to release dormancy is supported by: *ft1* mutant has high callose levels and unlike wild type, *ft1* mutant fails to reduce callose levels in response to cold, conversely, FT1 induction can downregulate callose levels even without cold. Moreover new data included in revised manuscript shows that FT1 induction can regulate expression of GA pathway (that regulates callose levels) as well as that of several PD related genes. As for translocation of FT1 in *abi1-1*, we have already shown earlier (Tylewicz et al., 2018) that buds of *abi1-1* mutant when grafted on FT1 overexpressors are able to initiate growth, whereas wild type buds or (*ft1* mutant buds) with closed PD are unable to initiate growth when grafted on FT1 overexpressors which strongly indicates the importance of FT1 mobility. As for spatial correlation between FT1 and callose, this is difficult to address as FT1 is already known to move as shown by us as well as Murata et al and thus could be both (in same as well as FT1 non-expressing cells).

8) Based on their results, the authors concluded that FT1 (together with GA) trigger dormancy break but their relationship is not further investigated. Enhanced expression of FT1 could be the effect and a consequence of callose degradation/reduced callose deposition at PDs and not the trigger of the response. Analysis of FT1 expression in the GA overexpression lines and analysis of dormancy release in GA20oxoOE *ft1* double mutant would clarify this aspect.

Response: Our expression data (see above showing GA20-oxidase and GA2-oxidase expression upon FT1 induction, now also added to the revised manuscript) shows that FT1 upregulates GA20-oxidase and suppresses GA2-oxidase. Thus our data shows FT1 is upstream of GA addressing the reviewer's comment on the relationship between FT1 and GA pathway.

As for enhanced expression of FT1 being a consequence of callose reduction is not supported by our results. For example, callose levels fail to be downregulated in *ft1* mutant whereas FT1 induction is sufficient to induce callose downregulation and moreover, FT1 induces GA pathway,

and also impacts the expression of several PD related genes as shown above (and included in the revised manuscript). Thus these data do not favor the scenario in which callose downregulation/PD opening induces FT1 expression. Instead, our data, together with those of others, support the scenario where FT1 downregulates callose rather than FT1 induction being a consequence of callose downregulation.

9) The authors here reported that level of expression of FT1 are lower in variable cold condition compared to cold condition; considering that the *ft1* mutant presented defects in dormancy break even in cold condition they concluded that warm spikes inhibit bud break. FT1 expression was quantified by qRT-PCR on the entire bud. Analysis of the pattern of expression of FT1 would determine whether the accumulation of callose is inhibited in the same cells where FT1 expression is triggered or if FT1 acts non-cell autonomously to determine dormancy break.

Response: We have tried to investigate FT1 expression *in vivo* by expressing FT1 its under native promoter and tagged with fluorescent protein. Despite several attempts we have been unable to express FT1 from its native promoter due to gene silencing of all the constructs from native promoter and therefore to visualize it in native conditions. However, we have shown earlier that FT1 movement is feasible and this requires open PD e. g. wild type dormant buds fail to reactivate when grafted on FT1 overexpressors whereas *abil-1* buds with open PD grafted on FT1 overexpressors can reactivate under the same conditions. Moreover, in-situ analysis by Andre et al shows that FT1 transcript is not expressed throughout the bud whereas our data shows that callose reduction is significant extensively throughout the bud. These results would suggest FT1 acts both in cells it is expressed but then FT1 movement also further amplifies callose reduction throughout the bud. Furthermore, FT1 induces GA pathway and GA itself moves via PD. Thus we feel that while knowing whether FT1 acts cell autonomously or non-cell autonomously is interesting, it is nevertheless not crucial to the key finding of the paper which is that buds decide to integrate cold by using warm spikes and that this is mediated by transcriptional regulation of FT1.

10) LIM1 regulation is one of the key responses to cold, and upregulation is reduced in variable cold conditions (Fig S4). could you provide some statistical analysis of these data? Are these qRT-PCR data? How would LIM1 mutants respond to variable cold, i.e., is LIM1 here the key factor that mediates the difference in response?

Response: As suggested by the reviewer, we have performed statistical analysis and as indicated earlier, the downregulation of LIM1 by warm spikes is not statistically significant. This now indicated in the revised manuscript. This would suggest that warm spikes do not act via LIM1 downregulation supporting our data.

11) The suggested bet-hedging strategy is certainly intriguing, but not really explored sufficiently in this manuscript. As stated above (point 9), I am missing the spatial component in the analysis of bud break. Are the breaking/non-breaking buds evenly distributed at the plant? How is the response of a bud, consisting of many hundred of cells, overall integrated? I am tempted to suggest to take this part of the story out of the manuscript, and explore this in-depth in a separate manuscript. It appears here like a very interesting, but insufficiently supported add-on.

Response: We thank the reviewer for positive response to our findings on variability. To clarify, in trees, lateral buds and apical buds behave differently in response to cold. Therefore, we have specifically considered analysis of apical buds for our experiments (also all other analysis was performed on apical buds). Which would mean single bud at the apex for every plant (n=20-25). Thus breaking/nonbreaking buds which would include analysis of both apical and lateral buds is not attempted in the context of this work. While we agree spatial analysis would be very useful, our findings are nevertheless highly interesting as indicated by the reviewer, to the tree community as they provide an insight into survival mechanism which has not been investigated so far. So we hope reviewer will allow us to keep this result. We are happy to incorporate the spatial aspect as suggested by the reviewer into the discussion (if necessary) as being important for the future and follow up to this work.

Reviewer #5 (Remarks to the Author)

This study provides important mechanistic insight into how plants integrate variable temperature signals to regulate bud dormancy release. The authors demonstrate that plasmodesmata-mediated cell–cell communication acts as a key integrator of fluctuating thermal cues, particularly through the detection and interpretation of intermittent warm spikes. They show that long warm periods suppress dormancy release by preventing PD reopening via repression of FT1 and GA20-oxidase expression. In contrast, shorter warm interruptions allow cold integration and promote dormancy release. These findings reveal a previously uncharacterized role for FT1 in callose downregulation and PD opening. The study also links temperature-responsive PD dynamics to phenotypic variability in bud break, suggesting a cellular basis for bet-hedging under unpredictable spring conditions. Overall, this work presents a compelling and original framework for understanding how plants maintain robust developmental timing amid environmental noise, with broader relevance to plant adaptation under climate change. The manuscript is well written, and the data presented are convincing, novel, and well suited for publication in Nature Communications. Below are a few comments and questions for the authors' consideration:

Response: We thank the reviewer for useful suggestions to improve our manuscript.

Comments:

Figure 1a and Figure 4a

The schematic suggests that warm spikes were applied at different times on consecutive days (e.g., from ZT20–ZT24 on the first day, and from ZT40–ZT44 on the second). Is this interpretation correct? If so, could the authors clarify the rationale for this design?

Response: This is misunderstanding. In both cases, the warm spikes were applied at the same time and is indicated in materials and methods. It only appears different due to schematic figure.

Figure 2, Figures S1 and S3 and discussion the observation of stronger callose deposition at PDs in the SAM compared to adjacent tissues raises an interesting possibility of cold-induced, cell–cell communication regulation. Do FT1 or GA biosynthesis genes exhibit SAM-specific expression

patterns under these conditions? Could this reflect SAM-specific expression of β -1,3-glucanases involved in PD opening?

Response: Perhaps there is a misunderstanding as we did not show data comparing callose levels in SAM with adjacent tissues. Nevertheless cold induced PD opening definitely favors the reactivation of cell-cell communication within the buds as is implied by our data and those of others. While we do not have exact cell resolution data, FT1 and GA biosynthesis genes are indeed expressed within the SAM (central bud) and this is true for β -1,3-glucanases.

Line 119 “In agreement with prior studies, callose levels decreased in dormant buds exposed constant cold (Fig. 2a, Fig. S1a, Fig. S2a).” Could the authors clarify how the measured callose levels were specifically linked to PD openness? This connection is not immediately evident from the provided images.

Response: At the tissue level, the quantification of callose levels shown in Fig. 2a and Fig. S1a (based on mean fluorescence intensity) and Fig. S2a (based on the number of gold particles per PD) reflects the well-established dynamic of high callose accumulation during dormancy and its reduction upon dormancy release.

After cold treatment, callose at PD decreased to levels normally seen in actively growing shoot apices, where PD are open. This decrease coincided with dormancy release and bud break. In contrast, callose remained high in buds exposed to fluctuating cold with warm spikes, and no bud break occurred. Reduced callose therefore serves as a proxy for increased PD openness, consistent with dormancy release.

Line 238 The data show that constitutive overexpression of GA20ox leads to bud break in nearly 50% of plants under variable cold conditions (Fig. S6). However, this phenotype does not fully phenocopy the effects of FT1 induction. Could the authors elaborate on potential reasons for this divergence? Does this suggest that FT1 may regulate additional GA-independent pathways required for PD opening and dormancy release?

Response:

Thanks for pointing this out. Indeed this result suggests that FT1 could additionally regulate a pathway that contributes to dormancy release. We have now stated this in revised discussion.

Line 292-297 if cold integration mechanism that enables robust sensing of autumn to late autumn/winter transition, as, typically, there is a shift towards progressively shorter periods of warm spikes relative to cold. A lack of response to cold when the warm spikes last longer can therefore ensure that cold registration is not initiated and chilling threshold is not achieved prematurely, preventing precocious dormancy release..... (Fig.5)

Response: We are not sure if there is a comment to be addressed?

Is callose deposition at PDs, and the resulting closure of cell–cell communication, only triggered when warm spikes become short enough? If so, at what point in this model does PD dynamics shift from closure to opening?"

Response: To clarify, PD are closed before buds are exposed to cold. Based on the experimental conditions using 2 different fluctuating conditions, our results would suggest that the switch from close to open is initiated when warm spikes are reduced from 4 to 2 hours i. e. between 4 and 2 hours. We have now also included new data as supp Fig. S13 (see below Fig. 1) that supports this by showing that in contrast with 4 hour warm spikes when there is no significant reduction of callose, callose is significantly reduced when warm spikes are reduced to 2 hours, the latter also triggering dormancy release.

Fig. 1 Callose quantification at PD in the shoot apical meristem of aspen buds after 11 weeks of short days and in response to 4 weeks of constant or variable cold (cold exposure followed by 2 hours at 20 °C) temperatures. (A) Graph showing callose quantification per PD based on mean signal intensity (each data point represents an individual PD). Quantification was performed on six sample surfaces per condition, each with an analyzed area of 7515 μm^2 . Statistical analysis was conducted using one-way ANOVA ($p < 0.0001$), followed by Tukey's test. Error bars represent the 95% confidence interval of the difference, and asterisks (****) denote a significant difference. (B) Callose immunofluorescence (purple) in semi-thin sections of aspen tissue visualized by confocal microscopy. Cell walls (grey) were stained with the fluorescent dye calcofluor. Callose accumulation (purple signal) was detected using an Alexa Fluor 555–conjugated secondary antibody. Scale bars: 10 μm .